# Benchmarking Encoder-Decoder Architectures for Biplanar X-ray to 3D Shape Reconstruction

**Mahesh Shakya**      **Bishesh Khanal**

Nepal Applied Mathematics and Informatics Institute for research (NAAMII)

{mahesh.shakya,bishesh.khanal}@naamii.org.np

## Abstract

Various deep learning models have been proposed for 3D bone shape reconstruction from two orthogonal (biplanar) X-ray images. However, it is unclear how these models compare against each other since they are evaluated on different anatomy, cohort and (often privately held) datasets. Moreover, the impact of the commonly optimized image-based segmentation metrics such as dice score on the estimation of clinical parameters relevant in 2D-3D bone shape reconstruction is not well known. To move closer toward clinical translation, we propose a benchmarking framework that evaluates tasks relevant to real-world clinical scenarios, including reconstruction of fractured bones, bones with implants, robustness to population shift, and error in estimating clinical parameters. Our open-source platform provides reference implementations of 8 models (many of whose implementations were not publicly available), APIs to easily collect and preprocess 6 public datasets, and the implementation of automatic clinical parameter and landmark extraction methods. We present an extensive evaluation of 8 2D-3D models on equal footing using 6 public datasets comprising images for four different anatomies. Our results show that attention-based methods that capture global spatial relationships tend to perform better across all anatomies and datasets; performance on clinically relevant subgroups may be overestimated without disaggregated reporting; ribs are substantially more difficult to reconstruct compared to femur, hip and spine; and the dice score improvement does not always bring a corresponding improvement in the automatic estimation of clinically relevant parameters. Code and pretrained-models are available at `https://github.com/naamiinepal/xrayto3D-benchmark`

## 1    Introduction

X-ray is the most common and widely used imaging modality for orthopaedics, trauma, and dentistry as it has low radiation, low cost, and is portable. X-ray scanner projects 3D information of the target body into a plane, resulting in a 2D image. This 2D representation is not ideal and sometimes not enough for visualizing the 3D structure that can be important in diagnosis, prognosis, surgery planning and navigation, and medical education. A CT scan captures X-ray-like images from several angles covering 360 degrees and reconstructs a single volumetric image, providing detailed 3D structural information of the target anatomy. However, CT scans have relatively high radiation, are costly, and are not even available in many rural health centres across the globe. Hence, there has been a longstanding interest in the scientific community to develop methods that can reconstruct 3D images or structures of interest from few to a single X-ray images of various human bones [25, 47, 50], teeth [49], and anatomies of other species [24].

From the early days of the stereo-correspondence point-based approach  [8, 41], several methods have been proposed for 3D reconstruction from biplanar radiographs: non-stereo-corresponding point-based, contour-based, statistical shape model (SSM) based, parametric, and hybrid methods [26].

37th Conference on Neural Information Processing Systems (NeurIPS 2023) Track on Datasets and Benchmarks.

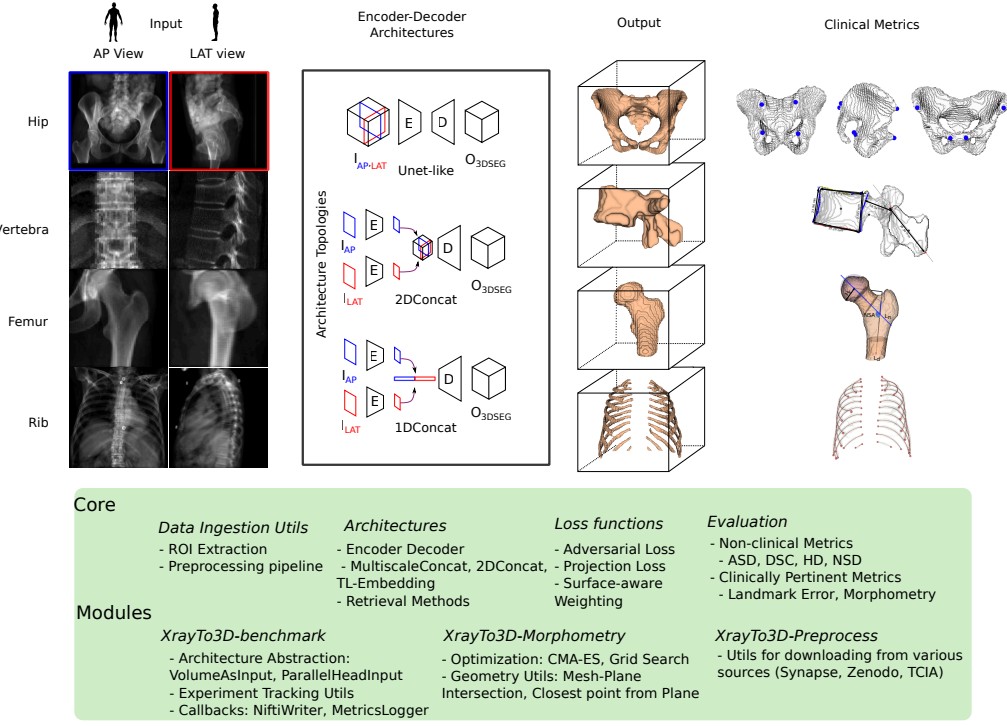

Figure 1: Benchmark Framework for Biplanar X-ray to 3D Shape Reconstruction

These methods face challenges in extracting hand-crafted features (landmarks or contours), in using deformation priors and atlas, or in building SSMs [3, 6, 28, 35]. Recently, several deep learning-based approaches have shown promise to provide more accurate results with faster computation time [33]. Although methods combining deep learning and anatomical priors via atlas or features such as landmarks or contours are starting to emerge [5, 12, 51], most of the existing deep-learning-based methods predominantly predict shape or image directly using an encoder-decoder-based neural network architecture [1, 4, 29, 38, 47].

Despite deep learning-based 2D-3D reconstruction from Biplanar X-ray being an active research topic with several different models proposed [1, 4, 29, 38, 47], we still do not know which methods are the best ones for what contexts. We observe four main challenges that need to be addressed for better progress in this field: i) *Lack of an open platform* that brings together publicly available dispersed datasets of different anatomies in different standards to a common standard; this makes it difficult for researchers developing new methods to evaluate their methods in a common benchmark. ii) *Lack of reproducible works* due to the use of private datasets, incomplete description of hyperparameters, and closed source code. iii) *Lack of a comprehensive evaluation* required to assess the potential of clinical translation: evaluation on single anatomy which is different for different papers; aggregated results without providing insight on the model's performance via subgroup analysis and failure modes.

iv) Limited effort towards *identifying and optimizing clinical parameters* that are of interest: most methods use metrics such as Dice Score or Hausdorff Distance for measuring reconstruction accuracy, but studies evaluating how these metrics impact the clinical parameters and decision-making are almost non-existent. For instance, it is unclear if the dice score of 0.8 instead of 0.9 would bring any difference to an actual clinical application where the method is applied.

In this work, we propose an open platform that brings together six dispersed and different multi-centre publicly available datasets into a common standard, and eight recently proposed deep learning models based on encoder-decoder architectures. We evaluate the models across four different anatomies, analyze results across multiple metrics disaggregated by various factors of variation that are of clinical interest, and assess their ability to estimate clinical parameters.

**Contributions** *Provide a comprehensive and standardized framework for datasets access, pre-processing, baseline comparison, and a set of metrics:* We provide an open-source implementation for accessing six different publicly available datasets, bring them into a common standard and pre-processing pipeline, provide reference implementations of SOTA architecture, analysis scripts for extracting clinically relevant parameters from reconstructed bone shape.

*Benchmark biplanar X-ray 2D-3D bone reconstruction methods:* We benchmark eight encoder-decoder based end-to-end architectures which were proposed specifically for X-ray to 3D bone shape reconstruction on four anatomies (vertebra, hip, rib and femur) using public datasets. Although we implement and evaluate encoder-decoder architecture, our benchmarking framework can be used for any type of 3D reconstruction model that takes biplanar X-rays as input and produces a 3D shape as a binary segmentation mask.

*Report disaggregated results across anatomical and pathological categories, dataset source etc. and highlight the limitations of the commonly followed approach of reporting aggregated results:* Specifically, we report disaggregated results on vertebra sub-types and pathologies.

*Robustness to Domain shifts:* We describe various benchmarking tasks to perform an external validation of the baselines in a realistic clinical context such as when the X-ray contains fractured bones or implants, or when the model is evaluated on an entirely different population cohort.

*Relationship of Dice score and error in the automatic estimation of clinically relevant parameter on 3D reconstructed results:* We implemented three algorithms to automatically estimate clinically relevant parameters from three different anatomies (7, 8, 9 parameters for vertebra, hip, femur respectively), and used it to study the relationship between dice score and the error in clinically relevant parameter estimation of 3D reconstructed shape against ground truth shape.

## 2 Related Work

Recent studies have highlighted the issues in deep learning research with too many architectures without a clear understanding of which one is better or disguising speculation as explanations [31], lack of reproducibility [7] and a gap in improvements in benchmarks vs. actual applications such as in clinical translations [52]. These studies have different scopes, such as the broad scope of machine learning in general but focusing on reproducibility [7] or field-specific contexts such as in health care covering broad issues [52]. Such recent works covering a broad field provide guidelines and emphasize important topics that need attention from the scientific community. Nevertheless, specific applications can have peculiar needs and issues that require further investigation and attention[13]. For example, the COVID-19 pandemic led to a very large number of papers on deep learning-based COVID-19 detection in a short period of time, but almost all these methods were clinically not useful and had study design flaws and bias [20, 43]. Benchmarking frameworks in other subdomains such as Uncertainty Quantification[37], Graph Neural Networks[16], Semi-Supervised Learning[54] etc. provide valuable insights and the platform to build upon for the scientific community. Our work is in the same direction and an attempt to move towards real clinical translation of deep learning based 2D-3D reconstruction methods.

## 3 Benchmarking Framework

To facilitate and unify the evaluation and comparison of encoder-decoder architectures, we propose a benchmarking toolkit providing scripts to automatically aggregate, curate, and preprocess datasets currently residing in varied data sources (CT scans and segmentation masks coming from different sources and originally released as medical image segmentation datasets). Reference implementations of the relevant encoder-decoder architectures are provided. Additionally, reference implementation of methods for obtaining clinically relevant metrics such as landmarks and morphometry are provided, going beyond generalized metrics such as Dice Score (DSC), Average Surface Distance (ASD), Hausdorff Distance (HD), Normalized Surface Distance (NSD).

### 3.1 Datasets

#### 3.1.1 Digitally Reconstructed Radiographs (DRRs) from CT scans as input images

The machine learning (ML) task in this work is to reconstruct the 3D structure of a target bone when a pair of Anterior-Posterior (AP) and Lateral (LAT) X-ray images of the target bone is given as input. Ideal training and evaluation of this task require manual ground truth segmentation of 3D structures in the CT scan of the same patient whose AP and LAT X-ray images are available as input. However, to our knowledge, there are no publicly available paired images of real X-ray scans and CT scans of the same patients. Since most methods construct AP and LAT Digitally Reconstructed Radiographs (DRRs) from CT scan images and use them as input to the model for training and evaluation, we use the same approach in this work.

#### 3.1.2 Data Ingestion: Four Different Bone Shapes from six Public CT Segmentation Datasets

Various CT Segmentation datasets are publicly available but are scattered across different data platforms. Additionally, these datasets require curation (in some cases manually) to obtain select anatomy-specific views. For example, TotalSegmentator[55] consists of multiple organ segmentations with heterogeneous views, which requires curating to select anatomy-specific views. In some cases, this can be done automatically, such as in the Rib Dataset where we reject CT scans with incomplete or absent ribs resulting in 481 usable data samples from 1204 available samples. In other cases, as in the Hip Dataset, this curation had to be done manually to avoid partial hip bones. CTSpine1K[14] and CTPelvic1K[32] provide manually annotated segmentation of spine and hip bones for publicly available CT scans from earlier other contributions from projects such as *MSD*([2], *KITS19*[23], *HNSCC*[19], *COLONOG*[27]. Pretrained CT Segmentation models[40] may also be used to obtain silver-standard ground truth[4].

We provide scripts to generate Biplanar Xray-3D segmentation dataset that streamlines this process. Our current implementation allows one to define a configuration file in the given format and run the scripts to generate the required datasets.

**TotalSegmentator-Rib-Dataset** 481 CT Scans were considered usable for generating Rib Dataset from the TotalSegmentator Dataset[55]. Samples containing the full set of ribs were only included.

**TotalSegmentator-Femur-Dataset** 465 CT Scans (of 1204 available) were selected from the TotalSegmentator Dataset[55] to be usable. Samples with only partial(45) or absent femur(676) were rejected based on voxel count (threshold set at 30k voxels). 41 scans contained only the left femur and 2 scans contained only the right femur.

**TotalSegmentator-Pelvic-Dataset** 446 CT Scans were considered usable for generating Pelvic Dataset. Samples with no Hip (572) bones and those with low voxel count (threshold set at 150k voxels) were rejected. Additionally, after visual assessment, 90 scans were rejected since they contained only partial sections of the hip. The *Identifier* of these rejected samples is provided in the Appendix and as part of the Benchmarking Framework Repository.

**CTPelvic1k-Pelvic-Dataset** CTPelvic1k [32] consists of 1106 pelvic CT scans with manual segmentation of left and right pelvic bones, sacrum, and vertebra near the sacrum. The dataset has seven subsets (number of scans in parenthesis): *Abdomen*(35), *Colonog* (714), *Msd-T10* (155), *Kits* (44), *Cervix* (41), *Clinic* (103), and *Clinic-metal* (14). *Clinic* consists of scans with fractured bones, and *Clinic-metal* has images having foreign bodies such as implants, screws, and rods. *Msd-T10* was unusable as it had highly anisotropic voxels which resulted in unrealistic and pixelated DRR.

**VerSe2019-Vertebra-Dataset** VerSe2019 [46] contains 160 CT scans with two types of annotations: manual segmentation of individual vertebra, and the centroid location provided as metadata. The vertebra with foreign materials such as cement and screws were removed(since there were too few such samples to learn from) resulting in 1722 vertebrae from the 160 CT scans.

**RSNACervicalFracture-Vertebra-Dataset** This dataset[44] consists of 710 vertebrae from 87 patients whose CT scans consist of mostly cervical vertebrae were selected after rejecting partial(212) vertebra.

### 3.2 Models

We evaluate representative architectures including Pure Transformer-based (Hatamizadeh et al. [21, 22]) and CNN-based (Bayat et al. [4], Chen and Fang [11], Girdhar et al. [18], Kasten et al. [29], Oktay et al. [39], Ying et al. [58]) encoder-decoders on four different bone anatomies representing a variety in shape, field-of-view, and surface complexity using publicly available datasets. The architectures proposed in the literature specifically for biplanar X-ray to 3D were originally validated as a proof-of-concept on single bone shape, many of them on single cohort and limited (< 30) test samples, and in some cases using a silver-standard dataset. This varied evaluation protocol means we cannot compare their performance based on the reported metrics. In this work, we evaluate these architectures under equal footing using a variety of datasets which include large multi-centre datasets.

#### 3.2.1 Encoder-Decoder Architectures

Encoder-Decoder-based Biplanar X-ray to 3D architecture consists of Encoder(s) that take in two X-ray images (AP and LAT view) and learn to encode statistically meaningful low-dimensional feature representation. The Decoder takes the low-dimensional 2D feature representation and learns 3D feature maps required to obtain the 3D bone surface segmentation. This distinct gap in dimensionality and semantics is overcome in architecture design via Two-Stage Method[18], or ii)End-to-End Learning[4, 11, 29, 58]. In the Two-Stage Method, an autoencoder learns the 3D bone shape manifold in the first stage. In the second stage, X-ray encoders are trained to map into the 3D bone shape manifold, followed by joint-tuning of this learned low-dimensional manifold. In the End-to-End Learning Method, the bottleneck layer at the interface between the last layer of the Encoder and the first layer of the Decoder reshapes the incoming 2D feature maps into pseudo-3d feature maps in addition to fusing the incoming 2D feature maps from the AP and LAT view Encoders along orthogonal axis[4, 29, 58]. This feature fusion can happen at various *scales* and *stages*. Transvert[4] fuses 2D feature maps at a single scale, whereas X2CT-GAN[58] and 3DReconNet[9] bridge 2D Encoder and 3D Decoder at multiple scales. 3DReconNet builds upon X2CT-GAN by using fully-convolutional architecture instead of a 1D bottleneck layer. Alternatively, instead of independently processing the AP and LAT views along two parallel Encoder branches, the two X-ray inputs may be fused along an orthogonal axis, resulting in a 2-channel 3D input volume. This allows popular state-of-the-art 3D segmentation architectures like UNet, and recent Attention-based architectures like SwinUNETR[21] to be used as-is.

*Hyperparameter Tuning:* All six architectures except Unet-like (UNet, AttentionUNet) have hyperparameters for depth and width dependent on input/output image size. As seen in Table 1, we define two volume sizes: one for hip, femur and rib (HFR), and another (smaller) for vertebra. Optimal depth-width parameters were manually optimized separately for vertebra and femur. The optimal depth-width parameters found for the femur were used as is for the hip and rib. Batch size and learning rate are kept the same across all datasets. The batch size (1 for SwinUNETR and UNETR, 4 for MultiScale2DConcat and 8 for others) is chosen to fit the larger model in the GPU (Titan RTX3090). The same learning rate (2e-4 for UNETR/SwinUNETR and 2e-3 for others) worked well for both depth-width parameters. For each dataset, all the architectures were trained for the same number of iterations, where the #iteration was chosen by manually checking when the slowest architecture converged to a stable loss (TotalSeg-Rib:4000, Verse19-Vertebra:15000, TotalSeg-Hip:3000, TotalSeg-Femur:4000).

### 3.3 Clinically Relevant Metrics

#### 3.3.1 Extracting clinically relevant Bone Morphometry Parameters from Segmentation

Relevant literature in Deep Learning based Biplanar X-ray to 3D Reconstruction generally rely on intrinsic evaluation using Image-based metrics rather than extrinsic evaluation based on downstream clinical tasks. Although designing specific clinical tasks and evaluating clinical decisions with approaches like a prospective clinical trial is not practical for studies focusing on developing methods, we believe that evaluating error in estimating clinically relevant parameters from reconstructed results can be an important step towards aiding clinical translation. While generalized metrics for segmentation such as Dice score (DSC), Hausdorff distance (HD), and Average Surface Distance(ASD) provide a useful measure, depending on the downstream task, certain anatomical metrics may be more (or less) important than others. For example, Chênes and Schmid [12] mention that although

the reconstruction error of the femoral head region of the proximal femur was the largest, this may not be an issue for clinical Total Hip Arthroplasty (THA) since the femoral head will be resected anyway.

**Femur Morphometry** We automatically extract Femoral Head Radius(FHR) and Neck Shaft Angle (NSA) from Femur Segmentation by adapting Cerveri et al. [10].

**Pelvic Landmarks** We implement a robust heuristic method described in Fischer et al. [17] to estimate landmark positions required to define the Anterior Pelvic Plane(APP) and Superior Inferior Spine Plane(SISP). These landmarks are used to study subject-specific biomechanics and implant design. We compare the distance between groundtruth landmarks and model-predicted landmarks as a clinically meaningful metric for reconstruction performance.

**Vertebra Morphometry** We use the method described in Di Angelo and Di Stefano [15] to obtain various vertebral measurements. Although the method was tested in the thoracic and lumbar vertebrae in the original work, we found that it also works well for the cervical vertebra. We tested the method on a much larger cohort than in the original work where the method was tested on 8 vertebra specimens. Although the method was robust at large, there were issues with classifying the vertebral body mesh boundary points in some of the cases. More work will be needed to improve robust automated vertebral measurements. Despite failures in some of the cases, we find that, at large, the method provides a useful tool to quantify model performance for clinically relevant metrics.

**Rib Morphometry** We extract morphometric parameters from individual ribs that assist in finite-elements modelling and estimate clinically meaningful parameters such as *rib area* whose overestimation is associated with overestimation of stiffness of thoracic spine construct[36]. The model prediction consists of many implausible rib shapes which means we cannot obtain clinically meaningful parameters and hence leave the automated extraction of clinical rib parameters from reconstructed rib shapes for future work.

# 4   Results

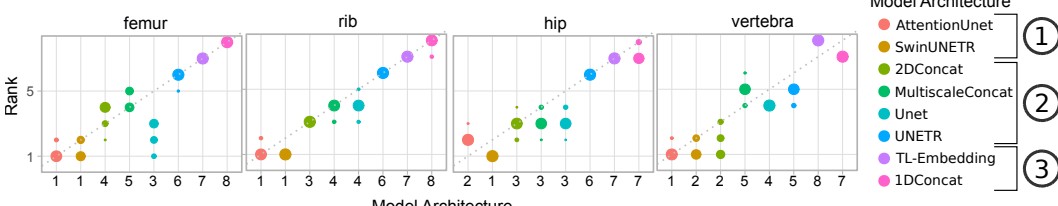

Figure 2: Ranking Stability by Task[56]: x-axis denotes each model architecture (with x-label denoting ranking) and y-axis denotes the ranking stability across various anatomy segmentation tasks. The blob plot is piecewise diagonal indicating that there is relatively clear ranking between architecture subgroups ①, ② and ③ whereas there is less stable separation within the subgroups.

As shown in Figure 2, across all tasks, we find that global spatial relationship learnt via self-attention and gating-based encoder-decoders such as SwinUNETR[21] and AttentionUNet[39] perform better (on average) than only learning a hierarchical local spatial relationship. However, it is important to note that AttentionUNet uses only 1.5 million parameters compared to 62.2 million of SwinUNETR.

Within the CNN-based Encoder-Decoders, there seems to be no clear winner among various architecture topologies categorized by feature-fusion depth and feature-fusion scales with 2DConcat[4], UNet[29] and Multiscale2D[58] performing comparably to each other. The architectures that destroy the spatial relationship at the bottleneck layer when learning low-dimensional embedding vector, namely 1DConcat[11] and TL-Embedding Network[18], perform worse among the compared architectures.

**Disaggregated Reporting of Metrics** Disaggregated reporting of metrics for the VerSe2019 dataset based on *level*, *shape* and *severity* in Figure 3 shows reduced performance on severely deformed vertebra but not as much on the mild and moderate deformity. Similarly, *crush*-shaped deformities are reconstructed poorly compared to other shape deformities (*wedge, biconcave*). These discrepancies are hidden when reporting metrics averaged over the whole dataset, resulting in misrepresentation of

Table 1: Benchmark Evaluation

| Dataset (#train/#test) (vol size) (voxel resolution) | Method Reference | #Param | Dice(%)↑ | HD95(mm)↓ | ASD(mm)↓ | NSD@1.5mm↑ |
|---|---|---|---|---|---|---|
| TotalSeg-Femur-DRR | SwinUNETR [21] | 62.2M | 93.64 | 3.32 | 0.94 | **0.85** |
| | AttentionUnet [39] | 1.5M | **93.66** | **3.12** | **0.93** | **0.85** |
| | 2DConcat [4] | 1.2M | 93.05 | 3.73 | 1.02 | 0.83 |
| | UNet [29] | 1.2M | 93.36 | 3.32 | 0.95 | 0.84 |
| | MultiScale2DConcat [58] | 3.5M | 92.79 | 3.64 | 1.11 | 0.81 |
| 786/138 | UNETR [22] | 96.2M | 92.39 | 3.96 | 1.16 | 0.80 |
| 128x128x128 | TL-Embedding [18] | 6.6M | 90.43 | 4.28 | 1.41 | 0.73 |
| 1.0 | 1DConcat [11] | 40.6M | 89.66 | 4.86 | 1.57 | 0.67 |
| TotalSeg-Rib-DRR | SwinUNETR | 62.2M | **54.05** | 6.17 | **0.90** | 0.49 |
| | AttentionUnet | 1.5M | 53.34 | **4.15** | 0.93 | **0.52** |
| | 2DConcat | 1.2M | 50.90 | 4.56 | 1.10 | 0.49 |
| | UNet | 1.2M | 49.10 | 4.86 | 1.35 | 0.48 |
| | MultiScale2DConcat | 3.5M | 49.23 | 7.29 | 1.12 | 0.47 |
| 408/73 | UNETR | 96.2M | 40.32 | 7.10 | 1.62 | 0.39 |
| 128x128x128 | TL-Embedding | 6.6M | 29.15 | 7.67 | 2.22 | 0.31 |
| 2.5 | 1DConcat | 40.6M | 27.22 | 16.98 | 2.60 | 0.28 |
| TotalSeg-Hip-DRR | SwinUNETR | 62.2M | **85.78** | **2.55** | 0.81 | **0.54** |
| | AttentionUnet | 1.5M | 85.03 | 2.60 | **0.78** | 0.52 |
| | 2DConcat | 1.2M | 84.75 | 2.69 | 0.81 | 0.49 |
| | UNet | 1.2M | 84.45 | 3.16 | 0.88 | 0.48 |
| | MultiScale2DConcat | 3.5M | 84.48 | 2.98 | 0.84 | 0.50 |
| 320/56 | UNETR | 96.2M | 82.27 | 3.35 | 0.97 | 0.44 |
| 128x128x128 | TL-Embedding | 6.6M | 79.33 | 3.87 | 1.12 | 0.38 |
| 2.25 | 1DConcat | 40.6M | 78.85 | 3.67 | 1.11 | 0.37 |
| VerSe19-Spine-DRR | SwinUNETR | 62.2M | 83.59 | 2.56 | 0.80 | **0.86** |
| | AttentionUnet | 1.5M | **83.66** | 2.43 | 0.74 | 0.85 |
| | 2DConcat | 1.2M | 83.62 | **2.35** | **0.72** | 0.85 |
| | UNet | 1.2M | 82.17 | 2.61 | 0.83 | 0.82 |
| | MultiScale2DConcat | 3.5M | 81.85 | 2.74 | 0.78 | 0.82 |
| 1451/271 | UNETR | 96.2M | 81.84 | 2.68 | 0.81 | 0.83 |
| 64x64x64 | TL-Embedding | 6.6M | 79.20 | 3.00 | 0.99 | 0.76 |
| 1.5 | 1DConcat | 40.6M | 80.92 | 2.76 | 0.83 | 0.80 |
| Aggregate | SwinUNETR | 62.2M | **79.27** | 3.65 | 0.86 | 0.68 |
| | AttentionUnet | 1.5M | 78.92 | **3.07** | **0.84** | **0.69** |
| | TwoDPermuteConcat | 1.2M | 78.08 | 3.33 | 0.91 | 0.67 |
| | UNet | 1.2M | 77.27 | 3.49 | 1.00 | 0.66 |
| | MultiScale2DPermuteConcat | 3.5M | 77.09 | 4.16 | 0.96 | 0.65 |
| | UNETR | 96.2M | 74.20 | 4.27 | 1.14 | 0.62 |
| | TLPredictor | 6.6M | 69.53 | 4.70 | 1.43 | 0.54 |
| | OneDConcat | 40.6M | 69.16 | 7.07 | 1.53 | 0.53 |

model performance of clinically relevant subgroups. In fact, good reconstruction performance on these minority subgroups, such as pathological cohorts, may be more valuable.

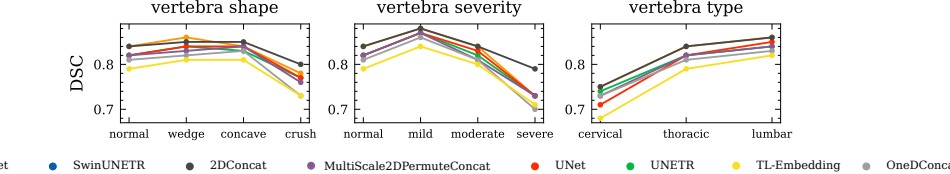

Figure 3: **Disaggregated metrics on VerSe19-Spine-DRR**: Reconstruction performance in clinically relevant subgroups that shows differing results for different subgroups.

**Reporting of Clinically Relevant Metrics** Figure 4 presents quantitative results for mean DSC vs. clinical parameter estimation error. One can see that the improvement in the mean DSC, averaged over the whole test set, does not always mean improvement in clinical metric estimation. For example, i) *first row*: in the fourth and fifth columns, OneDConcat has worse DSC by greater than five against UNet, but almost the same $PT_R$ and better $IS_L$. Similar results can be seen on the third from the right column when comparing OneDConcat against MultiScale2DPermuteConcat or SwinUNETR, ii) *second row*: Similar observation can be seen with DSC difference of greater than two for other pair of methods in column 3, 4, and 5, iii) *third row*: The third column has NSA not much correlated to DSC when DSC is higher than 90, and even +ve correlation between DSC vs. FDA estimation error in the third column from the right.

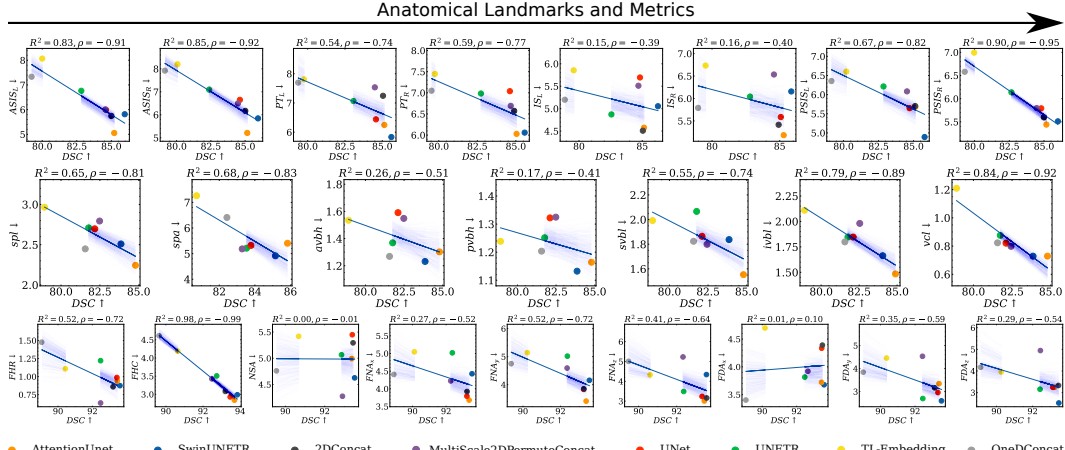

Figure 4: DSC vs Clinical Metrics: Top (hip), Middle (vertebra), Bottom (femur); The x-axis denotes the mean DSC obtained by each architecture whereas the y-axis denotes corresponding clinical metric estimation error. Improvement in mean DSC across the whole dataset does not always result in reduced clinical parameter estimation error, as we see in the figure for many parameters across architectures.

This effect is similar for other Image-based metrics such as ASD, HD95 and NSD.

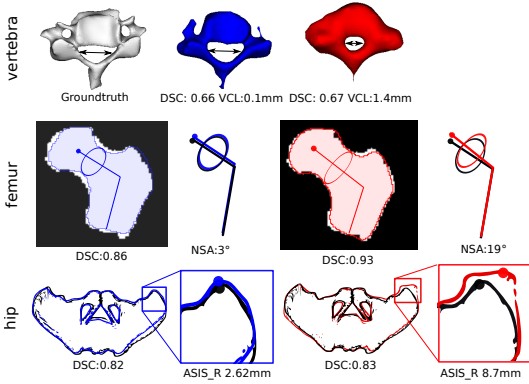

Figure 5: Qualitative examples where clinical metric and DSC diverge

In Figure 5, using sample results from our experiments, we present qualitative illustrations for three anatomies where DSC and clinical metrics are not consistent.

In the first row, the reconstruction with higher DSC is less anatomically plausible with higher Vertebra Canal Length (VCL) estimation error. This error can cause an overestimation of the safe zone for pedicle screw insertion resulting in incorrect surgical planning and outcome[30]. Similarly, in the second row, the Neck-Shaft Angle (NSA) is reconstructed well in the left example despite poor DSC compared to the one on the right. This signifies the need for measuring clinically relevant metrics, as NSA is an important characteristic of the proximal femur and useful in the diagnosis of various pathologies such as acetabular impingement, risk of hip fracture, hip dysplasia, and osteoarthritis [34, 42, 48]. Additionally, clinical metrics may be useful in quantifying reconstruction error in a localized region where DSC measures overall reconstruction accuracy. Case-in-point, in the last row of Figure 5, the Anterior Superior Illiac Spine (ASIS) is well reconstructed on the left with a large reconstruction error in the posterior ilium region whereas, on the right, the larger reconstruction error of the right ASIS, 8.7mm compared to 2.62mm, is compensated for by good reconstruction in other regions. These landmarks define points where muscles and ligaments attach, which may be used in personalized biomechanical evaluation[45] as well as for defining anatomical reference frame[57] useful for surgical planning, statistical modelling[53] etc.

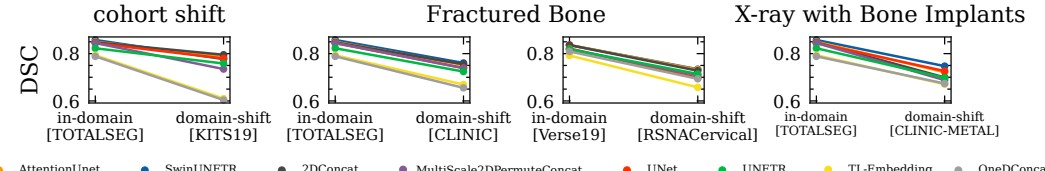

Figure 6: Reduction in performance due to Domain Shift: We find that shifts in the population cohort, as well as X-ray-specific shifts, hinder clinical translation.

**Domain Shift Results** Robustness to Domain shift is a challenge for clinical adoption. Figure 6 shows performance gaps of models on various groups of test images that have specific types of domain shift compared to the training set. *Kits19* subset of CTPelvic1k comes from different population cohorts and/or scanner-type. *Clinic* subset of CTPelvic1K contains images with hip fracture; *RSNACervical* contains images of the fractured cervical vertebra; *Clinic-metal* contains images with foreign implants such as cement, bone implants, screws, and rods. We observe that there is a substantial decrease in performance for all these new subsets. It is interesting to observe that while OneDConcat and TL-Embedding have lower DSC, their decrement in the score with domain shift is less severe than other methods in cohort shift and fractured bones.

**Misalignment Results**

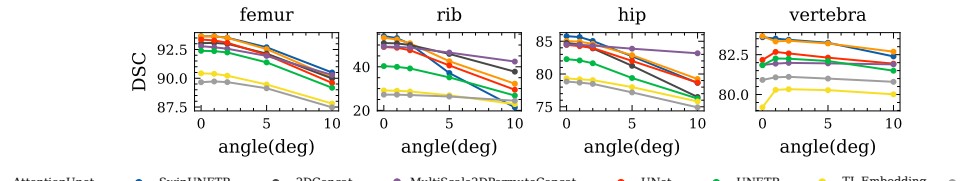

Figure 7: Reduction in performance due to misalignment: Misalignment due to perturbation of the relative angle of LAT view w.r.t AP view results in reduced performance for femur, rib and hip anatomies while vertebra-specific models are quite robust since such perturbed views occur naturally in the dataset.

Ideally, AP and LAT views are perfectly orthogonal. However, in practical clinical settings, misalignment errors can often result in slightly non-orthogonal AP and LAT views. We simulate such scenarios by generating misaligned DRR pairs of AP and Lateral views by generating LAT views at five different angles in the 91 to 100 degrees range. When evaluating the robustness of all the models to accurately reconstruct 3D when such misaligned data are presented during inference, we see (in Figure 7) that the models have degraded performance, which increases with the increase in misalignment. The reduction in DSC in the rib is most prominent with up to 30, while the performance is stable in the vertebra with delta DSC limited to 3. In the femur and hip delta DSC is moderate in the range of six and ten, respectively. Surprisingly, MultiScaleConcat[9, 58] architecture seems considerably more robust to such shift than others. The spine is a naturally curved structure with relatively bigger variation in the curvature in all three axes across the population. Thus, a more robust performance of the models in vertebra reconstruction may be because the models might already have learned to see the vertebra shape from diverse angles due to intrinsic natural variation across the population.

## 5 Discussion, Limitations, and Future Work

We introduced a novel open-source benchmarking platform for Biplanar X-rays to 3D bone reconstruction that – has a careful collection of six publicly available but scattered datasets on four anatomies; experimental designs to study versatility and robustness against clinically important subgroups and domain shift due to misalignment errors; provides a set of preprocessing and utility tools, and reference implementations of several reconstruction models with no public source code, reference implementation of several clinical metrics for three anatomies. We included only encoder-decoder architecture in this work, as they are the most widely used ones and there is a tradeoff in going for

exhaustive set of methods vs. performing large number of insightful experiments to build useful insights with limited resource and time budget. Nevertheless, our framework is not limited to the specific architecture, and we hope this work allows the community to compare newly proposed or other yet unimplemented models in a common set of clinically relevant benchmarking tasks across multiple anatomies. Using this benchmark, we evaluated task-specific as well as off-the-shelf architectures. Our results show that while existing reconstruction methods perform fairly well on aggregated data when using image-based evaluation metrics such as DSC. However, a closer look into disaggregated data show that the clinically important minority subgroups within datasets such as fractured bones or bones with implant have substantially reduced performance compared to commonly reported average scores. When analyzing the relationship between common image-based metrics to clinical metrics qualitatively and quantitatively, we find that improvement in metrics like DSC do not always lead to improved performance on clinical metrics.

When exploring several papers in the literature to implement clinical parameter estimation methods, we found that these methods were validated only on a small cohort with less diversity than the datasets on which we report our results. Despite challenges in automatically extracting clinical metrics on a large, diverse cohort, including model-predicted shapes that are not always anatomically plausible, the methods we implemented did reasonably well with only a handful of failure cases when visually going through the test dataset. Nevertheless, building more robust clinical parameter estimation methods might be an interesting avenue for the community to work on.

We use digitally reconstructed radiographs (DRRs) from publicly available CT scans only, but the real target clinical application would take during inference real X-rays pair. Hence, more accurate evaluations require several pairs of people's AP-LAT X-ray and CT scans, which are not readily available and are rarely used in the literature. The community's concerted efforts may help bring such datasets to public access. One of the very important clinical applications of 2D-3D reconstruction models is in resource-constrained settings such as large parts of LMICs in rural and community hospitals where CT Scans are unavailable or most people cannot afford the costly CT scans. However, the very use of the reconstruction would then be intended for diagnosing or planning surgery for abnormal bones. Thus, future work should focus on building robust methods that can reliably detect for various demographic subgroups and subgroups with specific abnormalities. Finally, those who are interested in prospective trials and evaluating the models effect on clinical decision making will likely need various clinical parameters estimated in this work. Thus, our work provides an important first step in consolidating the community's effort, and offering the ML researchers a useful platform and direction towards building 2D-3D reconstruction models that could be translated to the real world.

## 6 Acknowledgements

We would like to thank Dr. Bhaskar Raj Pant for providing the initial impetus for this project and convincing the team about the need for X-ray to 3D Reconstruction in trauma surgery for rural healthcare. This work was made possible via the institutional support by NAAMII Nepal. Additionally, We would like to thank our colleagues at NAAMII Nepal, especially TOGAI Team members - Kanchan, Prasiddha, Rabin, Safal and Manish, for being our sounding board to bounce ideas off and providing valuable technical support.

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
