# 1 Data Ingestion

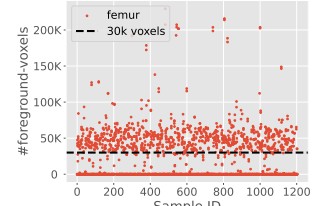
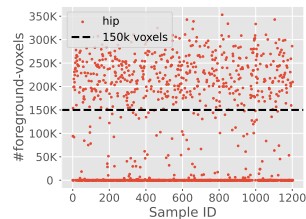

Figure 1: TotalSegmentator Dataset Ingestion: Selection of Samples was based on whether it contained a reasonable number of voxels (threshold defined individually for each anatomy) and then visually rejecting the remaining samples that contained only partial bones. Ribs were selected based on whether a full set of ribs were present.

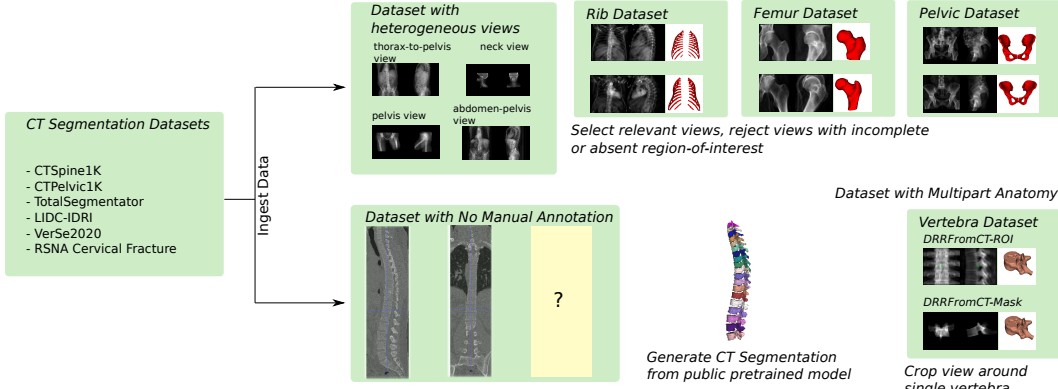

Figure 2: Data Ingestion: Various data preprocessing scenarios for Ingesting CT Segmentation Datasets for Biplanar X-ray to 3D Bone Shape Dataset

# 2 Benchmarking Tasks

Table 1: Benchmarking Tasks

| Benchmarking Evaluation Task | Training Dataset | Testing Dataset |
|---|---|---|
| Architecture Comparison | | |
|    Femur | TotalSeg-Femur | TotalSeg-Femur |
|    Hip | TotalSeg-Pelvic | TotalSeg-Pelvic |
|    Vertebra | Verse2019 | VerSe2019 |
|    Rib | TotalSeg-Ribs | TotalSeg-Ribs |
| Domain Shift: Fractured Bone | TotalSeg-Pelvic | CTPelvic1k-CLINIC |
| | Verse2019 | RSNA Cervical Fracture |
| Domain Shift: X-ray with Bone Implants | TotalSeg-Pelvic | CTPelvic1k-CLINIC-Metal |
| Domain Shift: Cohort Shift (Population, Scanner etc.) | TotalSeg-Pelvic | CTPelvic1k-KITS19 |
| Domain Shift: X-ray misalignment | TotalSeg-*, Verse2019 | TotalSeg-*,Verse2019 |

# 3 Hyperparameter Tuning

We split the dataset into train-val-tests by first splitting the whole dataset into the train-test split in the 85:15 ratio and then again splitting the train-split into the train-val split in the 85:15 ratio. We considered model selection for each of the models using the Dice Score metric on the train-val split. We used this validation performance to select the best hyperparameter setting and estimate the model training epochs. We then retrain the model using both train- and val-split as training data for

a fixed number of epochs determined during model selection and report metrics on test-split using the last epoch checkpoint. We choose the last epoch checkpoint since choosing the epoch with the best test-split metric would result in test-split leakage. We use default model sizes for off-the-shelf architectures such as AttentionUNet, UNETR and UNet.

| Method | Task | Encoder Channels | Kernel Size | lr |
|---|---|---|---|---|
| TLPredictor | femur | 8,16,32
**8,16,32,64**
16,32,64,128,256 | **3**,5 | **2e-3**
2e-4 |

| Method | Task | Encoder Channels | Decoder Channels | latent dim | kernel size | lr |
|---|---|---|---|---|---|---|
| AutoEncoder | rib | 4,8,16,32
**8,16,32,64**
8,16,32,64,128
16,32,64,128,256,128 | 4,8,16,32
**8,16,32,64**
8,16,32,64,128
16,32,64,128,256,128 | 64 | 3 | **2e-3**
2e-4 |

| Method | Task | Encoder Channels | Decoder Channels | fusion channels,depth | kernel size | lr |
|---|---|---|---|---|---|---|
| MultiScale2DConcat | femur | 4,8,16
8,16,32
4,8,16,32
8,16,32,64
**4,8,16,32,64,128** | 4,8,16
8,16,32
4,8,16,32
8,16,32,64
**4,8,16,32,64,128** | 32,2
32,3
32,4
32,5
**32,6** | 3 | 1e-2
2e-3 |

| Method | Task | Encoder Channels | Decoder channels | latent dim | kernel size | lr |
|---|---|---|---|---|---|---|
| 1DConcat | femur | **32,64,128,256**
32,64,128,256,512 | 128,1024,512,8,4,4,4
128,1024,512,256,128,64,32
**256,1024,512,256,128,64,32** | 128
**256** | **3**,5 | 2e-2 |

| Method | Task | Feature Size | num heads | dropout rate | lr |
|---|---|---|---|---|---|
| SwinUNETR | vertebra | 12
24
48 | 2,2,2,2
3,6,12,24 | 0.1 | 2e-3 |

| Method | Task | Encoder Channels | Decoder channels | kernel size | lr |
|---|---|---|---|---|---|
| AttentionUnet/Unet | | 8,16, 32, 64, 128 | 128,64,32,16,8 | 3 | 2e-2 |

Table 2: Model Architecture and Hyperparameter tuning configurations

# 4 Replicating reimplemented architectures

We performed an as-close-as-possible replication of Bayat et al and obtained comparable results (95.31% DSC in the original work vs. 94.43% in our replication), which is reasonable given that there is one important step without relevant information in the original paper that precludes exact replication. The training set in the original work was not the full set of images in the dataset, but an unknown subset. This is because the ground truth for the 3D reconstruction task in their case was a silver standard mask predicted by a deep learning segmentation model, and a radiologist manually went through the masks and removed 50 data points whose masks were deemed implausible. Original work reported results on the manually cleaned silver-standard dataset, but the information regarding the exact scans from the LIDC dataset that were discarded has not been made public.

For all other remaining architectures, the reported results are from private datasets. Some of our key motivations for this work are because of these challenges. Lack of reproducibility and disparate dataset quality makes it difficult for new methods to be compared with existing ones, which could potentially continue for newer methods in future if a common setting is not made available.

## 5   Benchmark Framework Usage

**Configuration File**

```yaml
---

# subject-list
subjects:
  subject_basepath: 2D-3D-Reconstruction-Datasets/lidc/subjectwise
  subject_list: configs/subjects_list/lidc_subject_list.lst

# xray image properties
xray_pose:
  _load: xray_pose_conf/${ROI_properties.axcode}_pose.yaml
  res: ${ROI_properties.res}
  size: ${ROI_properties.size}
  drr_from_ct_mask: ${ROI_properties.drr_from_ct_mask}
  drr_from_mask: ${ROI_properties.drr_from_mask}

# output directories
out_directories:
  _load: directory_conf/dir_ct.yaml

# ROI extraction properties
ROI_properties:
  axcode: PIR
  extraction_ratio:
    L: 0.5
    A: 0.5
    S: 0.5
  ct_padding: -1024
  seg_padding: 0
  drr_from_ct_mask: False
  drr_from_mask: False
  res: 1.0
  size: 96

# filename conventions
filename_convention:
  input:
    ct: "ct.nii.gz"
    seg: "seg.nii.gz"
  output:
    vert_xray_ap: "{id}_vert-{vert}_ap.png"
    vert_xray_lat: "{id}_vert-{vert}_lat.png"
    vert_centroid: "{id}_vert-{vert}_centroid.nii.gz"
    vert_centroid_xray_ap: "{id}_vert-{vert}_ap_centroid.png"
    vert_centroid_xray_lat: "{id}_vert-{vert}_lat_centroid.png"
    vert_ct: '{id}_vert-{vert}_ct.nii.gz' # add 'vert' for vertebra
    vert_seg: '{id}_vert-{vert}-seg-vert_msk.nii.gz'
    vert_overlay_ap: "{id}_vert-{vert}_ap_overlay.png"
    vert_overlay_lat: "{id}_vert-{vert}_lat_overlay.png"

```

## 6    Clinical Metrics

### 6.1    Vertebra Morphometry

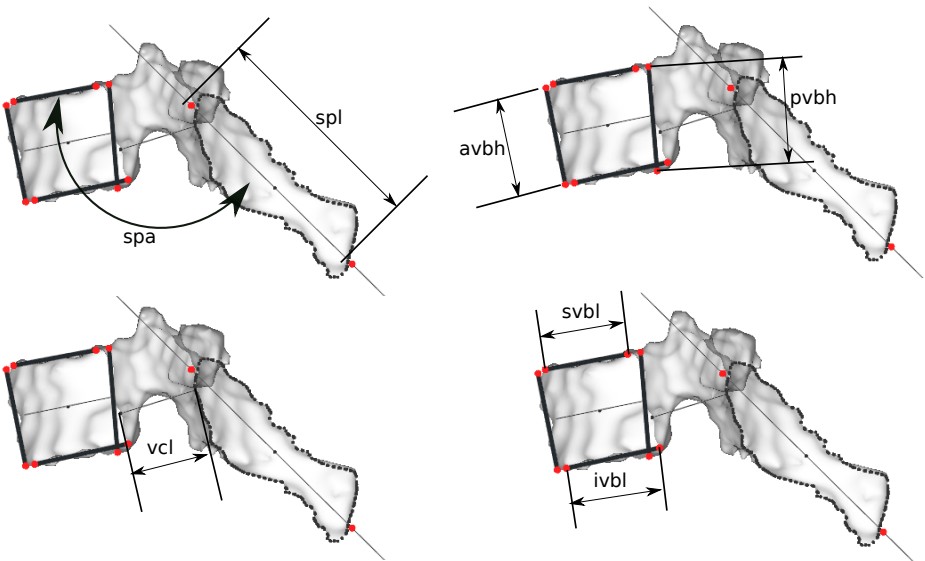

Figure 3: Vertebra Morphometry Metrics

**Femur Morphometry** We automatically extract Femoral Head Radius(FHR) and Neck Shaft Angle (NSA) from Femur Segmentation by adapting [**?** ]. The following adaptations were made: i) Since full-length femur bones were not available, automatic estimation of the diaphysis axis as described in [**?** ] was not possible. Hence, manual localization of the subtrochanteric region was performed on groundtruth segmentation and then transferred to predicted segmentation. This localization allows robust circle fitting to estimate the diaphysis axis. ii) Some of the samples do not even contain enough subtrochanteric region to reliably estimate the femur diaphysis axis. For these examples, Neck Shaft Angle(NSA) cannot be estimated. Additionally, [**?** ] requires estimation of the diaphysis axis for robust localization of the femoral head and neck region. As an alternative, for such cases, we transfer femoral head and neck localization from the groundtruth. The manual localization of the subtrochanteric region is provided in the Benchmarking Framework Repository.

We find that the variability due to these modifications is similar to the original method except for slightly increased variability in estimating (Femur Diaphysis Axis) FDA as shown in fig 4. We think that this ambiguity is due to not having enough subtrochanteric and diaphysis regions to accurately estimate FDA.

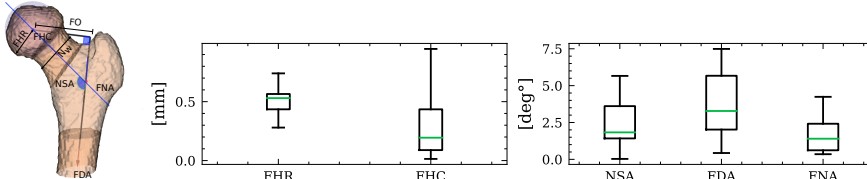

Figure 4: Repeatability of the femur morphometry extraction method as measured by error distributions for a) the landmarks/anatomical sizes and b) axis alignment identified by the adapted method.

| Method | In-domain [TOTALSEG] | OOD [KITS19] | Δ | OOD [CLINIC] | Δ | OOD [CLINIC-METAL] | Δ | In-domain [Verse19] | OOD [RSNA] | Δ |
|---|---|---|---|---|---|---|---|---|---|---|
| SwinUNETR | 85.78 | 77.68 | 8.09 | 76.04 | 9.74 | 74.71 | 11.06 | 83.59 | 73.42 | 10.18 |
| AttentionUnet | 85.03 | 78.64 | 6.39 | 75.22 | 9.81 | 72.14 | 12.89 | 83.66 | 73.23 | 10.43 |
| 2DConcat | 84.75 | 79.52 | 5.22 | 75.50 | 9.25 | 69.93 | 14.82 | 83.62 | 72.77 | 10.85 |
| UNet | 84.45 | 77.93 | 6.52 | 73.96 | 10.49 | 72.64 | 11.80 | 82.17 | 69.80 | 12.37 |
| MultiScale2DConcat | 84.48 | 73.48 | 11.00 | 73.83 | 10.65 | 68.79 | 15.69 | 81.85 | 70.83 | 11.03 |
| UNETR | 82.27 | 75.82 | 6.45 | 72.41 | 9.86 | 69.79 | 12.48 | 81.84 | 71.39 | 10.45 |
| TLPredictor | 79.33 | 61.00 | 18.33 | 66.92 | 12.41 | 67.07 | 12.26 | 79.20 | 65.74 | 13.46 |
| OneDConcat | 78.85 | 60.39 | 18.46 | 65.52 | 13.33 | 67.36 | 11.49 | 80.92 | 69.35 | 11.57 |

Table 3: Reduction in Performance due to Domain Shift: The reduction in DSC (represented by column Δ) when comparing In-Domain performance with Out-of-Domain(OOD) performance shows the need for robustness against relevant shifts for clinical acceptance.

# 7 Supplement to Quantitative Analysis of DSC vs clinical parameters

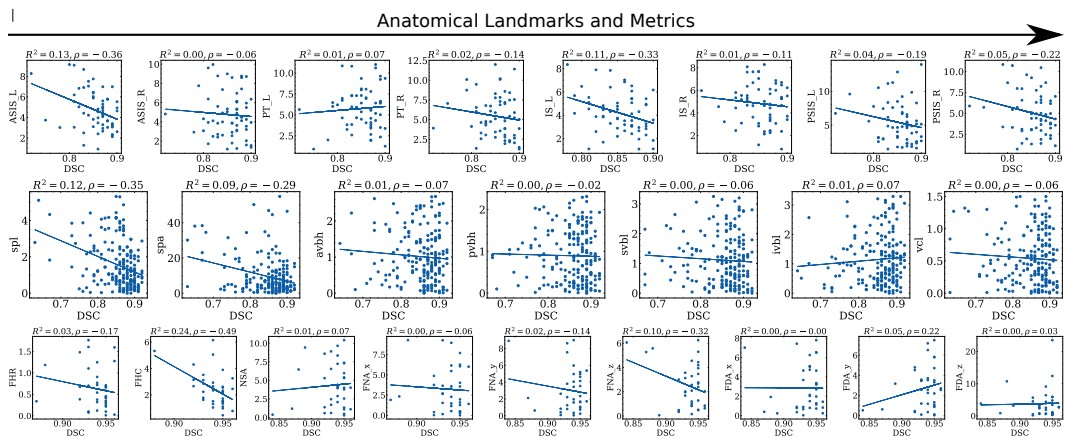

Figure 5: Relationship between Dice and Clinical Metrics across data samples on a single architecture(AttentionUnet): Top (Hip), Middle (Vertebra) and Bottom (femur)

 # 8   Qualitative Visualization

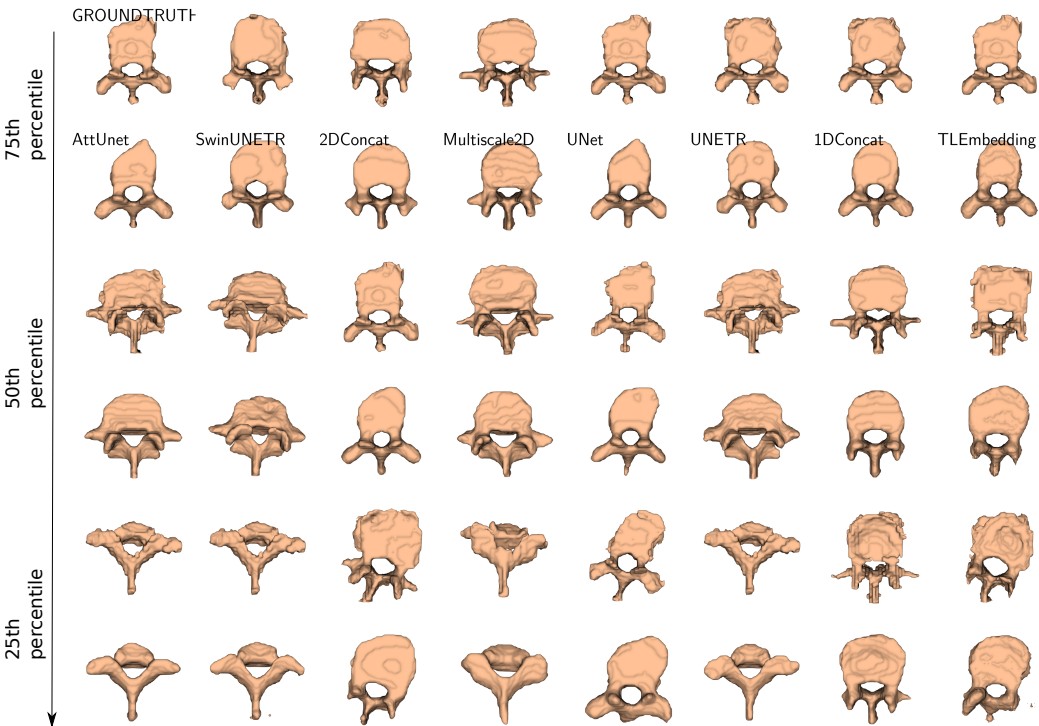

Figure 6: Vertebra Qualitative Results: 1st, 3rd and 5th row are groundtruth for corresponding architectures, whereas 2nd, 4th and 6th row are model predictions. The vertical axis represents the best(75th percentile), median and worse(25th percentile) samples for each architectures.

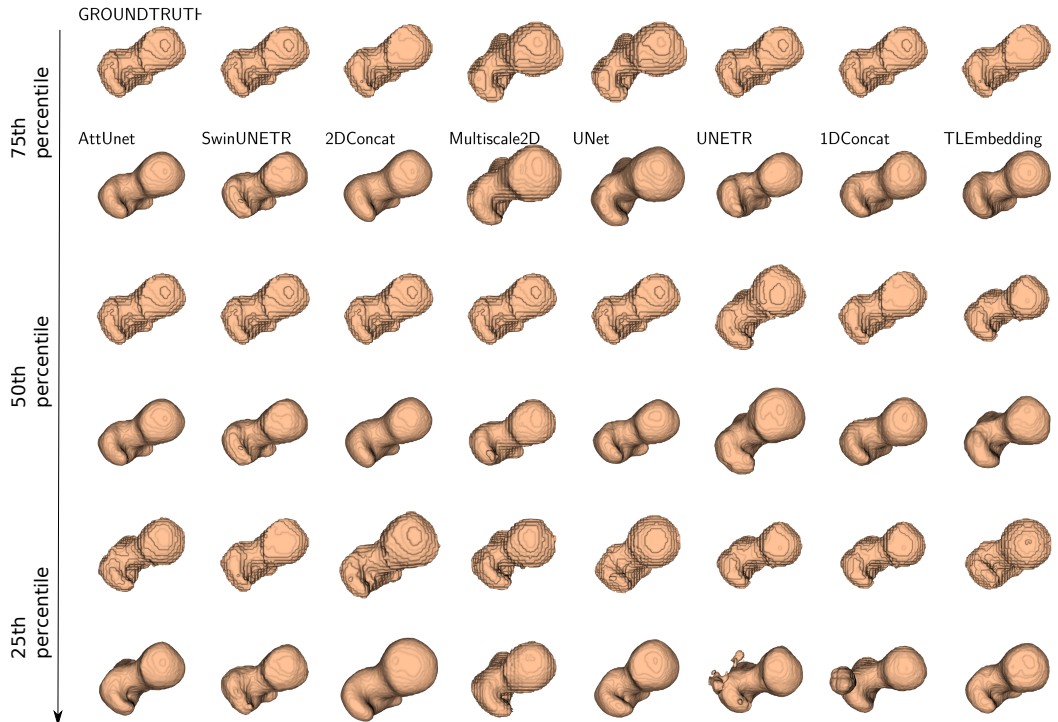

Figure 7: Femur Qualitative Results: 1st, 3rd and 5th row are groundtruth for corresponding architectures, whereas 2nd, 4th and 6th row are model predictions. The vertical axis represents the best(75th percentile), median and worse(25th percentile) samples for each architecture.

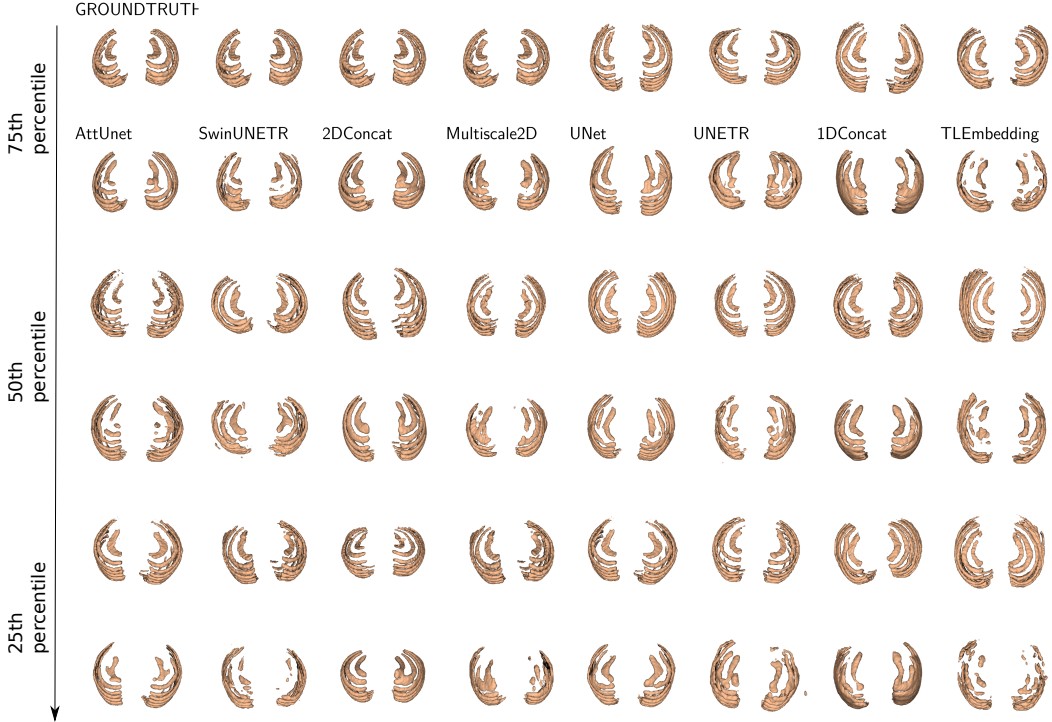

Figure 8: Rib Qualitative Results: 1st, 3rd and 5th row are groundtruth for corresponding architectures, whereas 2nd, 4th and 6th row are model predictions. The vertical axis represents the best(75th percentile), median and worse(25th percentile) samples for each architecture.

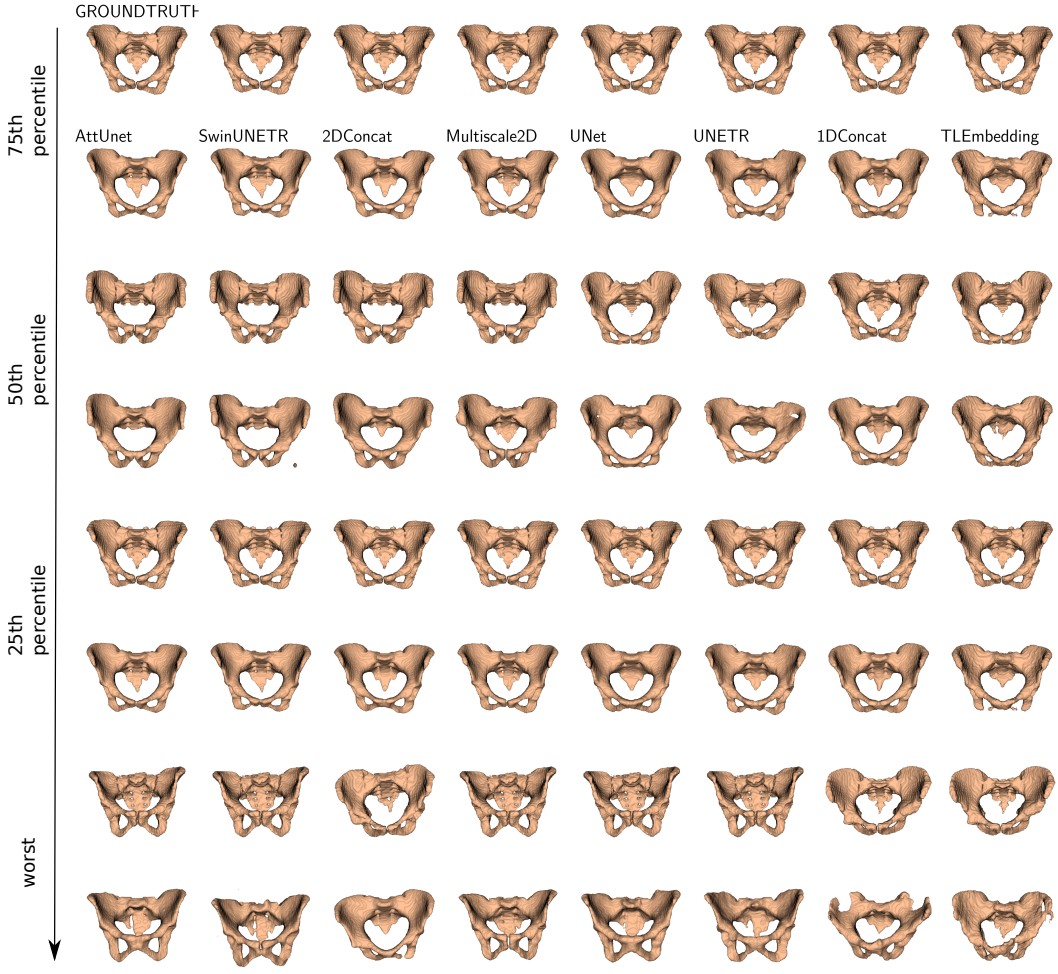

Figure 9: Hip Qualitative Results: 1st, 3rd and 5th row are groundtruth for corresponding architectures, whereas 2nd, 4th and 6th row are model predictions. The vertical axis represents the 75th percentile, median, 25th percentile and worse samples for each architecture.

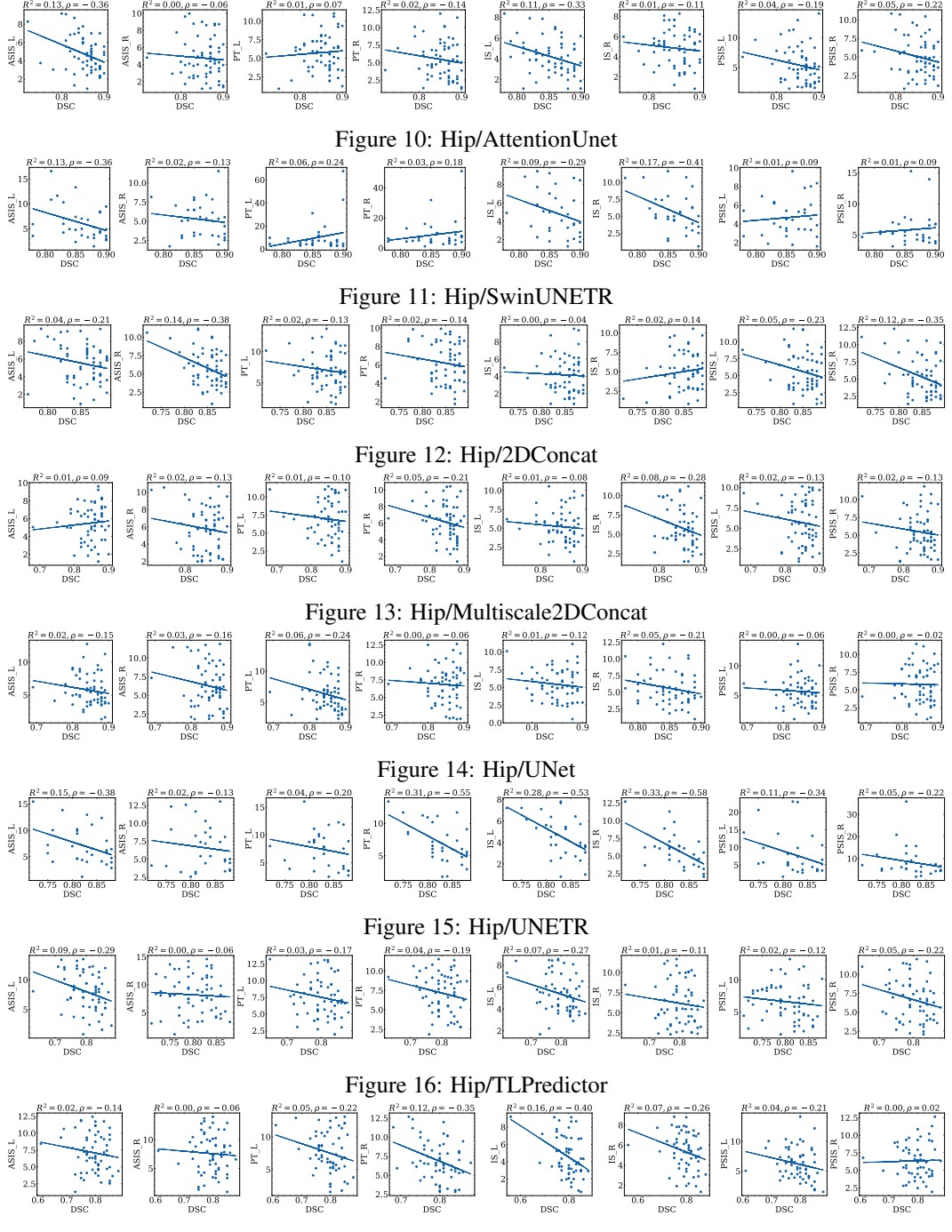

Figure 10: Hip/AttentionUnet

Figure 11: Hip/SwinUNETR

Figure 12: Hip/2DConcat

Figure 13: Hip/Multiscale2DConcat

Figure 14: Hip/UNet

Figure 15: Hip/UNETR

Figure 16: Hip/TLPredictor

Figure 17: Hip/1DConcat

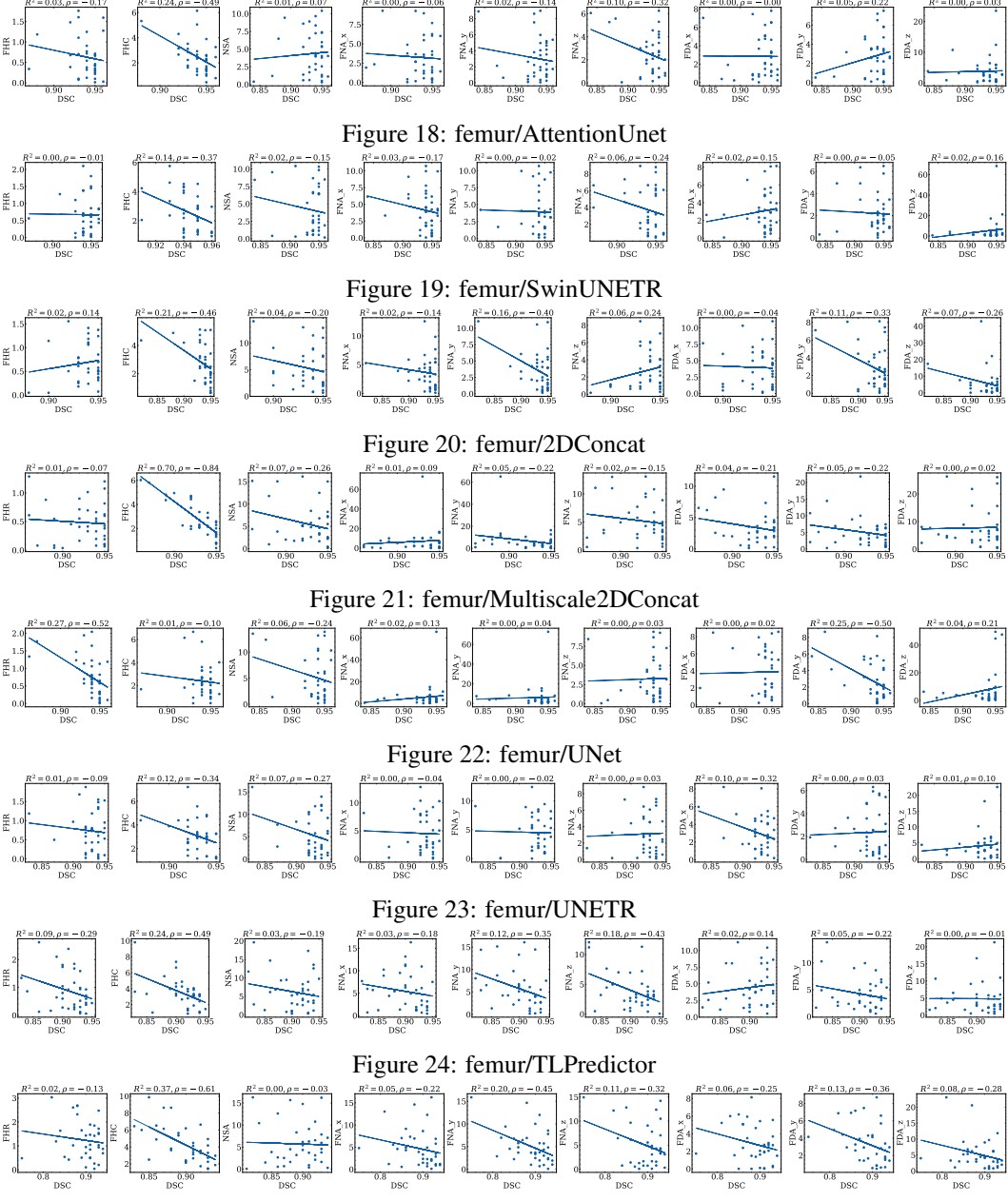

Figure 18: femur/AttentionUnet

Figure 19: femur/SwinUNETR

Figure 20: femur/2DConcat

Figure 21: femur/Multiscale2DConcat

Figure 22: femur/UNet

Figure 23: femur/UNETR

Figure 24: femur/TLPredictor

Figure 25: femur/1DConcat

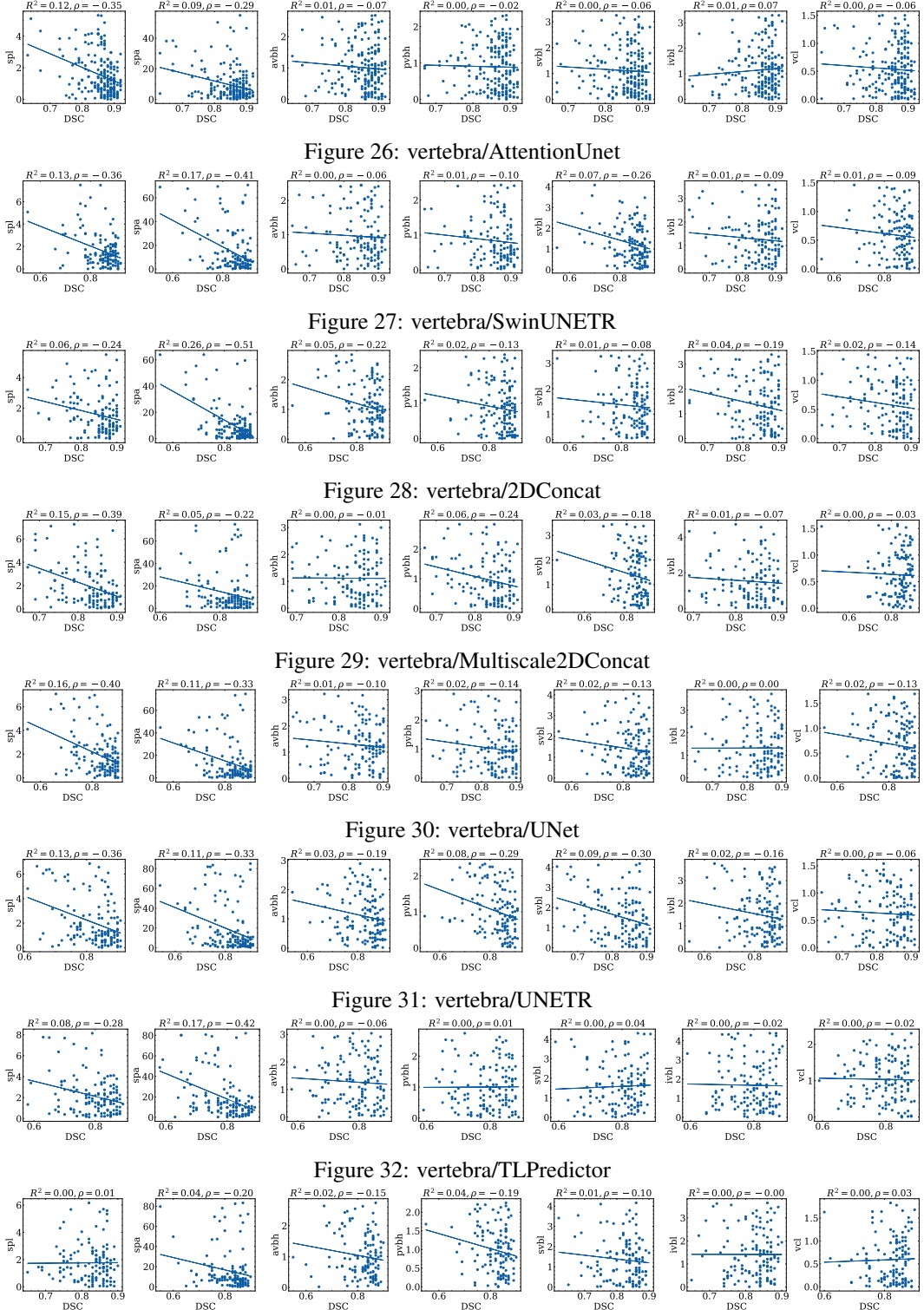

Figure 26: vertebra/AttentionUnet

Figure 27: vertebra/SwinUNETR

Figure 28: vertebra/2DConcat

Figure 29: vertebra/Multiscale2DConcat

Figure 30: vertebra/UNet

Figure 31: vertebra/UNETR

Figure 32: vertebra/TLPredictor

Figure 33: vertebra/1DConcat