# OpenReview forum: "Benchmarking Encoder-Decoder Architectures for Biplanar X-ray to 3D Bone Shape Reconstruction"
_NeurIPS.cc/2023/Track/Datasets_and_Benchmarks — NeurIPS 2023 Datasets and Benchmarks Poster_

### Official Review · Reviewer_HcgH · 2023-07-07
**A benchmark for 3D bone shape reconstruction from biplanar X-ray**

**Rating:** 6
**Confidence:** 3
**Clarity:** Yes, this paper is well organized.

**Strengths:**

The proposed benchmark holds promising potential for advancing the development of 3D bone shape reconstruction models from biplanar X-ray images in the future.

In their study, the researchers systematically categorized existing architectures and conducted a thorough comparison, encompassing recent vision transformer-based models among others. Their analysis provides a better understanding of the current state-of-the-art research.

The researchers further demonstrated their commitment by conducting extensive experiments and diligently highlighting the limitations present in existing works.

**Additional Feedback:**

(line 60, 61, 69) what does XX stand for?

**Correctness:**

The evaluation metric is clearly and precisely defined, providing a robust framework for assessing the performance of the models. Additionally, the construction of the dataset follows a reasonable approach, ensuring its relevance and suitability for the task at hand.

**Documentation:**

The paper lacks the URL to the dataset; however, the authors have explicitly stated their intention to release the repository containing the dataset.

**Limitations:**

While the AP and LAT views used in this study are generated from aligned CT scans, a valid concern arises regarding potential misalignments in clinical situations. It is crucial to evaluate whether the model can maintain its reconstruction performance when confronted with such misalignments.
Therefore, additional investigations or experiments could be conducted to evaluate the model's performance under different misalignment scenarios, simulating real clinical conditions. This would provide valuable insights into the model's limitations and enable researchers to devise strategies to improve its stability and accuracy when faced with misaligned data.

**Opportunities For Improvement:**

It is essential to investigate the reason behind the inconsistent performances of the OneDConcat model between Fig. 5 and Fig. 7. Notably, the performance of OneDConcat appears to be relatively low in Fig. 5, whereas it excels and achieves the best results in Fig. 7.

To provide a more comprehensive overview, it would be beneficial to present the average scores of each model in Table 1.

Regarding the visuals, the size of the dots in Fig. 4 and Fig. 7 appears to be too large, making it challenging to discern and accurately compare the scores. To improve clarity, it is recommended to resize the dots in the figures, ensuring better visual comparison of the model performances.

**Relation To Prior Work:**

Yes, they compared the dataset with existing datasets and explained the difference.

**Summary And Contributions:**

The paper introduces a benchmark for evaluating 3D bone shape reconstruction models from biplanar X-ray images. Various encoder-decoder architectures are evaluated with comprehensive experiments. The evaluation reveals the performance of various models across different anatomies and datasets, emphasizing the importance of attention-based methods and the limitations of relying solely on dice score for clinical parameter estimation.

---

> ### Author Response · Authors · 2023-08-27
>
> Thank you for the review and helpful feedback to improve the paper.
>
> > While the AP and LAT views used in this study are generated from aligned CT scans, a valid concern arises regarding potential misalignments in clinical situations. It is crucial to evaluate whether the model can maintain its reconstruction performance when confronted with such misalignments. Therefore, additional investigations or experiments could be conducted to evaluate the model's performance under different misalignment scenarios, simulating real clinical conditions. This would provide valuable insights into the model's limitations and enable researchers to devise strategies to improve its stability and accuracy when faced with misaligned data.
>
>
>
> We would like to thank the reviewer for suggesting an additional clinically relevant benchmarking task regarding potential misalignments between the AP and LAT view especially when the C-ARM does not support lateral head tilt in which case the patient has to be reoriented resulting in an inexact view or in bedside radiography requiring non-standard views such as Judet view [1]. We have added the aforementioned task as part of the Benchmark highlighting the clinical use case.
>
> We find all models have degraded performance which increases with the increase in misalignment (which is consistent with findings in the literature[2]) highlighting the model’s limitations. Surprisingly, MultiscaleConcat Architecture seems to be considerably more robust to such a shift than others.
> Future work may focus on building robust models for these transformations by learning the desired robustness via data augmentation or exploring techniques such as [3].
>
> Results
>
> https://anonymous.4open.science/r/xrayto3D-benchmark-ECF4/docs/images/hip.png
>
> https://anonymous.4open.science/r/xrayto3D-benchmark-ECF4/docs/images/rib.png
>
>
> *References*
>
> 1. Kovacevic, Djordje, and Arne Skretting. "Selecting the correct x-ray tube tilt angle and roof pillar rotation for bedside radiography with combined cranio-caudal and lateral cassette tilt." Radiography 14.2 (2008): 170-174.
>
> 2. Buttongkum, Danupong, et al. "3D reconstruction of proximal femoral fracture from biplanar radiographs with fractural representative learning." Scientific Reports 13.1 (2023): 455.
>
> 3. Saukh, Olga, et al. "Representing Input Transformations by Low-Dimensional Parameter Subspaces." arXiv preprint arXiv:2305.13536 (2023).
>
> >It is essential to investigate the reason behind the inconsistent performances of the OneDConcat model between Fig. 5 and Fig. 7. Notably, the performance of OneDConcat appears to be relatively low in Fig. 5, whereas it excels and achieves the best results in Fig. 7.
>
> We would like to thank the reviewer for pointing out the mislabeling(miscolouring) of the data in Figure 7 which makes it look like OneDConcat does very well wrt. other architectures.  We have fixed the typo and additionally added tables corresponding to the figures/graphs in the text in the supplementary section.
>
> >Regarding the visuals, the size of the dots in Fig. 4 and Fig. 7 appears to be too large, making it challenging to discern and accurately compare the scores. To improve clarity, it is recommended to resize the dots in the figures, ensuring better visual comparison of the model performances.
>
> We have incorporated the suggested dot size in the figure.
> Additionally, we would like to reiterate that our emphasis in Fig.4 and Fig.7 is to highlight the difference (reduction) in performance between subgroups when disaggregated reporting and show that such a trend is true for all the architectures.
>
> >To provide a more comprehensive overview, it would be beneficial to present the average scores of each model in Table 1.
>
> We have added an additional row to Table 1 presenting the average scores of each model aggregated across reconstruction tasks.

---

### Official Review · Reviewer_EdSm · 2023-07-20
**Review for #415**

**Rating:** 6
**Confidence:** 5
**Correctness:** I did not notice any errors in the da…

**Strengths:**

-the use of out-of-domain evaluation (training on same anatomy but with a different CT data source),
-many (xx) different algorithms and sub datasets used, and hence the generalisation seems good
-the authors make a good attempt to choose a metric/metric that is/are more clinically relevant
-They analyse ranking stability, and perform sub-grouping of different algorithmic concepts

**Additional Feedback:**

After reading the other reviews and the very thoughtful rebuttal along with the revised paper I am now slightly more inclined towards recommending acceptance (I increased my score by one notch). The authors released anonymised source code and aimed to address many concerns and in particular provided a better description, reasoning and discussion of the clinical metrics. Some parts remain ambiguous but overall this paper makes a nice contribution. If clinical metrics are automatically computed in a differentiable way they could even be used to complement/replace current loss functions in image reconstruction and segmentation.

**Clarity:**

The description is overall clear, however, due to page limitations some details are only found in supplementary material of references.

**Documentation:**

The appendix details the hyperparameter tuning a bit better, but the GitHub repository has not been provided. I therefore could not access relevant details about the processing pipeline.

**Limitations:**

Major: abstract claims up-to-date benchmark relied only on Dice, surface distances etc. and clinical metrics are necessary, however 4.3. states that “improvement in mean dice reported over a dataset when comparing one architecture over the other generally results in an improvement in mean clinical metric estimation.” Cf also Fig. 5. So there seems to be a lesser problem with previous sota.  Fig.6 that uses disaggregated scores now yields different results by comparing individual data points. However, I strongly believe that comparing two different 3D segmentation methods (either two manual/automatic ones, or a mix) based on those automatic anatomical metrics could yield only weak correlation, which would point to the problem in deriving these metrics.

-Not really a new dataset and limited methodological variation for selected methods, e.g. what about segmenting the 2D projections views first with a network and concatenating those outputs as well
-Poor results on Rib dataset not really explained.
-Why is computational demand not considered of relevance: Att.-UNet requires 40x less Paramentes than SwinUNETR

**Opportunities For Improvement:**

-The writing is not flawless yet. Some important details have simply been forgotten to be inserted in the final version (red xxs), in general the grammar is not always correct in terms of using plurals, the/a ..
-The contributions part is not completely written yet (last item misses the descriptive part)
-I would suggest to reproduce partial results for re-implemented techniques on the published datasets to ensure correctness. “varied evaluation protocol means we cannot compare their performance based on the reported metric” does not prevent one from comparing the used model on their originally published data.
-No or unclear hyperparameter tuning (only some information in appendix) could lead to incomparable results. Which dataset (the same?) was used to adapt parameters?
-Another issue: The resolution of the TotalSegmentator dataset (originally 1.5mm)  is adapted and for some datasets lowered to generate a VOI that corresponds with the input size, however for  some applications a higher resolution could be desirable.
To address the problematic results of the rib dataset: Judging from the provided qualitative reconstruction results, it could be that the lower resolution of the input images make it more difficult to generate complete but delimited rib shapes. It would be interesting to see whether an approach operating on smaller patches of the thorax in combination with a sliding window would lead to improved results.

**Relation To Prior Work:**

The clinical metrics that are at the core of the proposed benchmark are well referenced but seem to have been problematic to use: „all of these methods fail for some of the examples in our benchmark dataset“. Maybe a more thorough literature research could reveal alternative strategies that work better and would strengthen the hypothesis that Dice/Hausdorff are less informative than those clinical measures.

**Summary And Contributions:**

The work compiles a new dataset of 3D segmentations with corresponding DRRs (digital reconstruction radiographs) and compares several published benchmark algorithm on 2D-3D reconstruction performance. One key contribution is to evaluate the methods with both segmentation-based metrics (Dice/Hausdorff) and clinical landmarks etc.

---

> ### Author Response · Authors · 2023-08-27
> **General reply: contributions of the work**
>
> We thank the reviewer for very helpful comments that have made us realize that we were perhaps a bit unclear in highlighting our contributions, and did not offer a better insight into the relationship between dice score and clinical parameters.
>
> We would like to emphasize that, in addition to this work being a compilation of datasets of 3D segmentation with corresponding DRRs and comparing published methods, as mentioned in the common reply to all reviewers, this work offers a set of tools and reference implementations together with comprehensive evaluation designs that are important if we want to translate the ML methods to real-world. Out of five key architectures published in the literature for this task, only one had publicly available source code (MultiScale2DConcat (Buttongkum et al.)), and another only one had used publicly accessible data to evaluate the proposed model (2DConcat(Bayat et. al)). We implemented and are providing open source the reference implementation of all these five architectures (+ adding three SOTA transformer-based backbones to one of the architectures which were not previously proposed in the literature, resulting in eight methods). Since all other proposed architectures used different privately owned datasets (mostly in individual anatomy), it was impossible to reproduce those results and unfortunately no benchmark existed that studied these methods in common settings across multiple anatomies. Our work presents the first open-source benchmark platform offering a wide range of bone anatomy through curated publicly available datasets, reference implementation, and comprehensive study to analyse important aspects for clinical translation: domain adaptation, disaggregated reporting across bone types and pathologies (often missed in original methods), and study of the relationship between dice scores and clinical parameters.
>
> In the responses that follow, we will address the specific points.

---

> > ### Comment · Reviewer_EdSm · 2023-08-29
> > **Will revised paper be uploaded during rebuttal?**
> >
> > I would like to sincerely thank the authors for making great efforts to address the various concerns from all reviewers. I think the answers do in part alleviate some issues. However, it would be nice to see the proposed changes in a revised manuscript (for other papers this was done, so I assume it is easily possible). Furthermore, could the authors please briefly explain whether the full source code is now released through the anonymous Github code that was linked in some comments?

---

> > > ### Author Response · Authors · 2023-08-31
> > > **Paper uploaded and addressing additional issues**
> > >
> > > Thanks once again for the very useful review that improved our paper.
> > >
> > > We have uploaded the revised version of the paper.
> > >
> > > The source code is indeed available as it is intended to be open source and publicly available.
> > > It can be accessed at [this link](https://anonymous.4open.science/r/xrayto3D-benchmark-ECF4/README.md), the link will be updated in the paper after the decision with a deanonymized Github repository URL.
> > >
> > > We respond to some final remaining comments below:
> > >
> > > > No or unclear hyperparameter tuning (only some information in the appendix) could lead to incomparable results. Which dataset (the same?) was used to adapt parameters?
> > >
> > > We have added additional details in the Benchmarking framework section of the revised version of the paper, part of which is copied below:
> > >
> > > “All six architectures except Unet-like (UNet, AttentionUNet) have hyperparameters for depth and width dependent on input/output image size. As seen in Table 1, we define two volume sizes: one for hip, femur and rib (HFR), and another (smaller) for vertebra. Optimal depth-width parameters were manually optimized separately for vertebra and femur. The optimal depth-width parameters found for the femur were used as is for the hip and rib. Batch size and learning rate are kept the same across all datasets. The batch size (1 for SwinUNETR and UNETR, 4 for MultiScale2DConcat and 8 for others) is chosen to fit the larger model in the GPU (Titan RTX3090). The same learning rate (2e-4 for UNETR/SwinUNETR and 2e-3 for others) worked well for both depth-width parameters. For each dataset, all the architectures were trained for the same number of iterations, where the \#iteration was chosen by manually checking when the slowest architecture converged to a stable loss (TotalSeg-Rib:4000, Verse19-Vertebra:15000, TotalSeg-Hip:3000, TotalSeg-Femur:4000). "
> > >
> > >
> > > >Poor results on Rib dataset not really explained.
> > >
> > > Although the exact reasons for poor results need further investigations, we do have some speculation and educated guess on why this might have happened. Various characteristics of the Rib dataset separates it from other anatomy datasets. One, the ratio of background / foreground ratio.  {"hip": 719, "femur": 612, "vertebra": 23, "ribs": 5231, "rib": 5231}. Another, unique topology of the rib: long thin slender. The cross-section diameter of a rib can be as low as 8 mm. Finally, for a single rib bone projected in X-ray image plane, there are many other interfering rays from other rib bones that gets projected to the same place. This makes the surface blurry in the 2D image, while in 3D the bones have distinct surface and separated by narrow space.
> > >
> > >  > Another issue: The resolution of the TotalSegmentator dataset (originally 1.5mm) is adapted and for some datasets lowered to generate a VOI that corresponds with the input size, however for some applications a higher resolution could be desirable. To address the problematic results of the rib dataset: Judging from the provided qualitative reconstruction results, it could be that the lower resolution of the input images makes it more difficult to generate complete but delimited rib shapes. It would be interesting to see whether an approach operating on smaller patches of the thorax in combination with a sliding window would lead to improved results.
> > >
> > > Thanks for the interesting suggestion. We tried that experiment by taking higher resolution rib images at 1.25 mm (increased from 2.5mm), taking 8 patches to train, and aggregating results on smaller patches with a sliding window approach. However, we found that it does not really improve the scores as seen below.
> > >
> > >
> > >  Method | #Params | DSC | ASD | HD95 | NSD
> > > |:------:|:--------------------:|:-------------:|:--------:|:------------:|:--------:|
> > > AttentionUnet|  1.5M|  49.70|   11.93 | 2.13 |  0.62
> > > UNet|  1.2M|  45.57|   9.88 | 2.01 |  0.59
> > > MultiScale2DPermuteConcat|  3.5M|  38.58|   27.95|  2.76|   0.45  |
> > > OneDConcat|  40.6M|  24.63|   36.29  |4.50 |  0.34

---

> ### Author Response · Authors · 2023-08-27
>
> >  abstract claims up-to-date benchmark relied only on Dice, surface distances etc. and clinical metrics are necessary, however, 4.3. states that “improvement in mean dice reported over a dataset when comparing one architecture over the other generally results in an improvement in mean clinical metric estimation.” Cf also Fig. 5. So there seems to be a lesser problem with the previous sota. Fig.6 which uses disaggregated scores now yields different results by comparing individual data points. However, I strongly believe that comparing two different 3D segmentation methods (either two manual/automatic ones, or a mix) based on those automatic anatomical metrics could yield only a weak correlation, which would point to the problem in deriving these metrics.
>
>
> We would like to kindly reiterate that, there are no existing “up-to-date benchmark” relying on DICE or related scores at all. Our work presents the first benchmark study, even when using the DICE and related scores only, for 2D biplanar X-rays to 3D bone reconstruction task.
>
> We have realized that our initial explanation of Figure 5 does not provide the full picture. The improvement of the mean dice score across a whole dataset for a particular method has in many cases not provided improvement but rather worsening of the clinical parameter estimation. As the question is related to a few other questions from other reviewers and needs a more comprehensive answer, we are responding jointly in the reply titled, “Study of DICE score (DSC) and clinical parameters relationship”.
>
> > Why is computational demand not considered relevant: Att.-UNet requires 40x fewer Parameters than SwinUNETR
>
> The computational demand is indeed relevant and this was the reason we put the number of parameters in the results. We thank the reviewer for the comment that helped us realize that we missed discussing this point in the main text. We are now adding the computational demand to our discussion in the revised version.
>
> In short: “Attention UNet performed almost as well as the best method SwinUNETR, despite having 40X fewer parameters. Moreover, for many clinical parameters, DSC increments do not necessarily bring substantial improvement in the parameter estimation. Hence, when selecting 2D-3D reconstruction methods or trying to propose new ones, it might be worth looking at the improvement in clinical parameters too instead of just looking at dice score vs. computational demand. For those who are building new methods, our framework facilitates exploring such questions allowing the community to focus on the core task of architecture design without spending too much time replicating and comparing with baseline architectures as well as robust evaluation.”

---

> ### Author Response · Authors · 2023-08-27
> **Reproducing partial results for reimplemented techniques**
>
> We have now performed an as-close-as-possible replication of Bayat et al and obtained comparable results (95.31% DSC in the original work vs. 94.43% in our replication), which is reasonable given that there is one important step without relevant information in the original paper that precludes exact replication. The training set in the original work was not the full set of images in the dataset, but an unknown subset. This is because the ground truth for the 3D reconstruction task in their case was a silver standard mask predicted by a deep learning segmentation model, and a radiologist manually went through the masks and removed  50 data points whose masks were deemed implausible. Original work reported results on the manually cleaned silver-standard dataset, but the information regarding the exact scans from the LIDC dataset that were discarded has not been made public.
>
> For all other remaining architectures, the reported results are from private datasets.
>
> |Architecture| 		Anatomy 	|Dataset	|
> |:------:|:------:|:------:|
> UNet	[1]		|Knee|		Private
> TL-Embedding[2]	|Wrist	|	Private
> OneDConcat[3]	|Vertebra	|Private
>
> Some of our key motivations for this work are because of these challenges. Lack of reproducibility and disparate dataset quality makes it difficult for new methods to be compared with existing ones, which could potentially continue for newer methods in future if a common setting is not made available.
>
> 1. Kasten, Yoni, Daniel Doktofsky, and Ilya Kovler. "End-to-end convolutional neural network for 3D reconstruction of knee bones from bi-planar X-ray images." Machine Learning for Medical Image Reconstruction: Third International Workshop, MLMIR 2020, Held in Conjunction with MICCAI 2020, Lima, Peru, October 8, 2020, Proceedings 3. Springer International Publishing, 2020.
>
> 2. Shiode, Ryoya, et al. "2D–3D reconstruction of distal forearm bone from actual X-ray images of the wrist using convolutional neural networks." Scientific Reports 11.1 (2021): 15249.
>
> 3. Chen, Chih-Chia, and Yu-Hua Fang. "Using bi-planar x-ray images to reconstruct the spine structure by the convolution neural network." Future Trends in Biomedical and Health Informatics and Cybersecurity in Medical Devices: Proceedings of the International Conference on Biomedical and Health Informatics, ICBHI 2019, 17-20 April 2019, Taipei, Taiwan. Springer International Publishing, 2020.

---

> ### Author Response · Authors · 2023-08-27
>
> > The clinical metrics that are at the core of the proposed benchmark are well referenced but seem to have been problematic to use: „all of these methods fail for some of the examples in our benchmark dataset“. Maybe a more thorough literature research could reveal alternative strategies that work better and would strengthen the hypothesis that Dice/Hausdorff are less informative than those clinical measures.
>
> Reference implementations of automatically estimating clinical parameters and their use are indeed one of the many other important contributions of the paper. While we explored several if not exhaustive sets of papers in the literature when implementing the algorithms, it is important to note that the methods presented in the literature were mostly validated on a small cohort with less diversity than the datasets on which we report our results. It is actually a positive outcome or takeaway of our work that has highlighted the gap in the literature and introduced it to the ML community that – for evaluating reconstruction tasks’ ability to reconstruct clinically relevant substructures on a wide range of datasets and pathology, there is a need for more robust automatic estimation methods. We believe that even with a more exhaustive search of implemented estimation methods, one would hardly achieve perfectly accurate results for all the complex scenarios present in ground truth and predicted reconstruction (we give some examples below on the challenges). Hence, we resorted to reasonably suitable methods and then focused on the core contributions rather than going into the comprehensive evaluation of parameter estimation methods which is out of the scope of current work.
>
> Moreover, in order to illustrate our point on why metrics like DSC do not always provide the full picture, we have now added some qualitative results (please see common reply titled, “Study of Dice score (DSC) and clinical parameters relationship”).

---

### Official Review · Reviewer_bLVp · 2023-07-21
**Benchmarking Encoder-Decoder Architectures for Biplanar X-ray to 3D Shape Reconstruction**

**Rating:** 8
**Confidence:** 4
**Correctness:** Yes

**Strengths:**

"Complete" package - i.e. curated data, scripts to extract same data, models, metrics and benchmarks.

Clinical metrics - this is interesting and important, as they have implemented standardised, clinically relevant metrics, that the ML community should be more aware of.

**Additional Feedback:**

None

**Clarity:**

Generally yes.

The paper has a few red XX marks, indicating something needs fixing or filling out. I'm not sure what these mean, or if its for anonymisation, but it doesn't appear to affect much.

**Documentation:**

The authors haven't provided a link to a repo, so I can't really assess this.

Supplementary information suggestion licensing is suitable, so I have no ethical concerns.



**Ethics:**

No issues that I can see.

**Limitations:**

None as such. No ethical concerns.

**Opportunities For Improvement:**

Mainly points of clarity:

1. In Figure 1, in the green box, the top row of features is useful, but the bottom row "API" is a bit too vague. What is the benefit to the reader? I think it needs more explanation.

2. In Figure 2: Im not sure this adds anything, or is of any use to the reader, and if anything the left hand side is more confusing to me than the diagram in Figure 1.

3. In Figure 5: I don't understand it. The caption says "Improvement in Dice", but I can't see what m_1 and m_2 refer to. I can see that as the DICE increases, clinical error measures generally go down, but I can't see the measure of \delta DICE. I think you are just plotting each model, so the x-axis is just DICE, not \delta DICE.

4. Figure 6 could do with more discussion. You are saying that when different models generally give better dice, you also generally get better clinical scores (Figure 5), but looking at individual samples, better DICE does not mean better clinical scores (Figure 6)? Ideally, we'd need to know how much an error in reconstruction affects the measurement error on the clinical scores and what the tolerance of measurement is before it becomes clinically unusable. Can you provide any more insight here?

**Relation To Prior Work:**

Yes

**Summary And Contributions:**

The authors present an open platform containing data, models and scripts to evaluate AI algorithms for biplanar X-ray (2D) to 3D shape reconstruction. The authors claim 5 main contributions - (1) a platform, (2) benchmark, (3) disaggregated results giving more insight into model performance on different anatomy/pathology, (4) evaluation against domain shift and (5) evaluation of DICE score against clinical metrics. All of these are important for the community, and the overall contribution is significant.

---

> ### Author Response · Authors · 2023-08-27
>
> Thank you for the review and helpful feedback to improve the paper.
>
> **Figure 1 Update**
>
>  > In Figure 1, in the green box, the top row of features is useful, but the bottom row "API" is a bit too vague. What is the benefit to the reader? I think it needs more explanation.
>
> Thank you for the suggestion. We have updated the “API” row (renamed as modules) to highlight the structure and features provided by i) the benchmark framework to assist in abstracting various architecture implementations, experiment tracking, and hyperparameter tuning ii) the morphometry framework providing utilities for geometric operations, optimization strategies such as CMA-ES, Grid Search etc. and iii) the preprocessing framework providing utilities for downloading datasets from various sources (Synapse, Zenodo, TCIA archive),
>
> [Updated Diagram](https://anonymous.4open.science/r/xrayto3D-benchmark-ECF4/docs/images/abstract_diagram_updated.pdf)
>
>
>
> **Figure 2 Update**
>
> > In Figure 2: I'm not sure this adds anything or is of any use to the reader, and if anything the left-hand side is more confusing to me than the diagram in Figure 1.
>
> Thanks for pointing this out. We agree that Figure 1 already captures most of the information that are important in the main paper. We will move the second figure to supplementary material, and use the saved space later in the paper to provide some qualitative illustration of the results and some clinical metrics.
>
> **Figures 5 and 6 comments on DSC vs. clinical metrics**
>
> *About Figure 5*: Thanks for pointing this out. Yes, the x-axis is just DSC and the dots are the mean dice score of a method over individual datasets. We have updated the caption to reflect this and removed unclear/unnecessary notations.
>
> *About the relationship between DSC and clinical metrics*: Yes, we have realized that Figures 5, and 6 and the study of the relationship between DSC and clinical parameters all need better presentation from our submitted version. Even in Figure 5, when we said that better DSC, in general, gives better estimation, it does not give the full picture as we can see many parameters where a better mean DSC score across the whole dataset does not improve the clinical parameter estimation. As the question is related to a few other questions from other reviewers and needs a more comprehensive answer, we are responding jointly in the reply titled, “Study of DICE score (DSC) and clinical parameters relationship”.

---

> > ### Comment · Reviewer_bLVp · 2023-08-29
> >
> > Thanks. I think you've done a good job responding to comments. I'll leave my score as-is, as I think it's fair, and positive in the first place.

---

### Official Review · Reviewer_KPqo · 2023-07-21
**A benchmarking framework for 3D bone shape reconstruction from biplanar X-ray images**

**Rating:** 6
**Confidence:** 3
**Clarity:** The paper is generally well-written a…

**Strengths:**

* The authors propose a comprehensive benchmarking framework that evaluates tasks relevant to real-world clinical scenarios. It allows for a standardized comparison of different models.
* The authors have made their platform open-source, which will be a valuable resource for other researchers in the field.
* The paper presents an extensive evaluation of eight 2D-3D models using six public datasets, which adds to the robustness of their findings.
* The authors have focused on tasks and metrics that are directly relevant to clinical applications, highlighting the potential for their work to have a real-world impact.


**Additional Feedback:**

No

**Correctness:**

The method used for constructing the benchmarking framework is sound, and the evaluation methods and experiment design are appropriate and performed correctly.

**Documentation:**

The authors have not explicitly provided code to reproduce the work.

**Limitations:**


* The authors have addressed some limitations of their work.

**Opportunities For Improvement:**

* Provide more details about the process of defining a configuration file and running the scripts to generate the required datasets.
* Discuss more concretely the clinical implications of the work.
* Provide a more detailed discussion of the limitations of the proposed benchmarking framework.
* The novelty of this work is weak.

**Relation To Prior Work:**

The authors provide a good review of the relevant literature and clearly explain the methods they used.

**Summary And Contributions:**

This study introduces a comprehensive benchmarking framework designed to assess encoder-decoder models employed in the 3D reconstruction of bone shapes from biplanar X-ray images. The authors offer an open-source platform that includes reference implementations of eight distinct models, thereby facilitating replication and further development by other researchers. The construction of the benchmarking framework is based on a solid methodology, and the evaluation techniques and experimental design have been appropriately and accurately executed.

---

> ### Author Response · Authors · 2023-08-31
>
> Thank you for the review and helpful feedback to improve the paper.
>
> >Provide more details about the process of defining a configuration file and running the scripts to generate the required datasets.
>
> We provide instructions in the software documentation of the open-source repository currently available at this link https://anonymous.4open.science/r/xrayto3D-benchmark-ECF4
>
> >Discuss more concretely the clinical implications of the work.
>
> Thank you for the comments. We have now uploaded a revised version of the paper that discusses more concretely the clinical implications of the work. The new presentation on the DSC vs. clinical parameters study [posted as a reply https://openreview.net/forum?id=NoE8g3LRAM&noteId=wHzcz8EDQG] and new material in the Results section provide some concrete examples. Similarly, the final section provides more implications, part of which is copied below:
>
> “.... One of the very important clinical applications of 2D-3D reconstruction models is in resource-constrained settings such as large parts of LMICs in rural and community hospitals where CT Scans are unavailable or most people cannot afford the costly CT scans. However, the very use of the reconstruction would then be intended for diagnosing or planning surgery for abnormal bones. Thus, future work should focus on building robust methods that can reliably detect for various demographic subgroups and subgroups with specific abnormalities. Finally,
> those who are interested in prospective trials and evaluating the model's effect on clinical decision making will likely need various clinical parameters estimated in this work. Thus, our work provides an important first step in consolidating the community’s effort, and offering the ML researchers a useful platform and direction towards building 2D-3D reconstruction models that could be translated to the real world.”
>
> >Provide a more detailed discussion of the limitations of the proposed benchmarking framework.
>
> We have now expanded our discussion on the limitations in the last section; the major parts related to limitations are:
>
> “We included only encoder-decoder architecture in this work, as they are the most widely used ones and there is a tradeoff in going for exhaustive set of methods vs. performing large number of insightful experiments to build useful insights with limited resource and time budget.”
>
> “When exploring several papers in the literature to implement clinical parameter estimation methods, we found that these methods were validated only on a small cohort with less diversity than the datasets on which we report our results. Despite challenges in automatically extracting clinical metrics on a large, diverse cohort, including model-predicted shapes that are not always anatomically plausible, the methods we implemented did reasonably well with only a handful of failure cases when visually going through the test dataset. Nevertheless, building more robust clinical parameter estimation methods might be an interesting avenue for the community to work on.”
> “We use digitally reconstructed radiographs (DRRs) from publicly available CT scans only, but the real target clinical application would take during inference real X-rays pair. Hence, more accurate evaluations require several pairs of people's AP-LAT X-ray and CT scans, which are not readily available and are rarely used in the literature.”

---

### Official Review · Reviewer_x1j9 · 2023-07-28
**A useful step toward a benchmark for a specific task of interest to the biomedical community**

**Rating:** 6
**Confidence:** 3

**Strengths:**

Recently, a number of datasets have been made available that cover a wide range of interesting problems. It is wholly useful for domain experts to comb through these datasets and reassemble them such that they are more amenable to training and evaluation. It is also desirable to think critically about the metrics we use to evaluate these methods. On both counts, the manuscript does a good job of providing tools to the biomedical research community interested in developing new methods and establishing new modes of evaluation. For these reasons, it seems well suited to the current venue.

**Additional Feedback:**

It's not clear to me why line 61: "We evaluate the models across different XX anatomy, analyze results across multiple metrics disaggregated by various factors of variation that are of clinical interest, and assess their ability to estimate clinical parameters"
is anonymized in this fashion.

**Clarity:**

While I feel the manuscript would benefit a great deal from further development of the Discussion/Conclusion, on the whole, the paper is well written and easy to follow. It well lays out the challenges and a strategy to solve it.

**Correctness:**

It is difficult to evaluate this with no visibility into the data or assembly framework. On its face, the approach and claims appear to be correct.

**Documentation:**

It is not possible to judge this based on the material provided. The authors have not yet released the Github repositories.

**Ethics:**

While there are obvious concerns any time patient data is used, this study does not provide any new data. Data is selected and processed from existing, publicly available datasets. Therefore, I do not suspect any ethical concerns with the submission that warrant further review.

**Limitations:**

The benchmark bills itself as specific to encoder-decoder architectures. This feels oddly specific and leaves the reader wondering why that is the narrow focus here rather than other types of models for 3D reconstruction. The manuscript would benefit from a few sentences stating why that is the case.

**Opportunities For Improvement:**

While this represents a valuable first step in many ways, it is very much a first step. For instance, clinical metrics are touted in several areas of the manuscript but it appears that much of this has been left for future work - i.e. "work towards (sp) identifying protocols and steps".

**Relation To Prior Work:**

The authors have well placed the work within the field, highlighting the shortcomings of multiple new models with limited means of benchmarking them against each other.

**Summary And Contributions:**

The authors have selected specific examples from 6 publicly available datasets to collate a baseline to assist in clinically-relevant evaluation of encoder-decoder architectures for 3D shape reconstruction of biplanar X-rays. Ostensibly, they have provided a framework to extract relevant data to recreate their analysis. Additionally, they use the resulting data set to benchmark several encoder-decoder based architectures using multiple disaggregated metrics. They have included evaluation against traditional metrics, like Dice score, in addition to clinically relevant validation criteria.

---

> ### Author Response · Authors · 2023-08-27
>
> Thank you for the review and helpful feedback to improve the paper.
>
>
> **Beyond Encoder-Decoder Architectures**
>
> Our benchmarking framework itself can be used by any type of 3D reconstruction model that takes biplanar X-rays as input and produces a 3D shape in the form of a binary segmentation mask. The implementations we chose to incorporate at this point of time were limited to encoder-decoder architectures which are the most widely used and recent architectures. However, our framework allows one to easily add more models of new types that can be evaluated and compared to existing models implemented in the framework.
> We will edit the manuscript to clarify this point, that our framework per se is not limited to encoder-decoder architectures.
>
>
> **Further development of Discussion/Conclusion**
>
> We are expanding the discussion and conclusion section and will be available in the revised version of the paper that we will upload soon.
>
> **Typo and minor comments about XX**
>
> We have corrected the typo and updated it to reflect four anatomies across six datasets, with eight different methods from five architectures. For the UNet architecture, we add three new methods where we replace the originally proposed backbone with three recent transformer-based backbones, which provide SOTA results.

---

> > ### Author Response · Authors · 2023-08-31
> >
> > Thanks once again for the helpful feedback that helped to improve the paper.
> >
> > - The source code is available at [this link](https://anonymous.4open.science/r/xrayto3D-benchmark-ECF4)
> >
> > - We have revised the presentation on clinical metrics in the revised paper, also available as a common reply to all the reviewers titled, [Study of Dice score (DSC) and clinical parameters relationship](https://openreview.net/forum?id=NoE8g3LRAM&noteId=BgzoJT4L07)
> >
> > - We have expanded the discussion and conclusion in the revised version of the paper that is uploaded.

---

### Author Response · Authors · 2023-08-27
**Thank you to reviewers**

We thank all the reviewers for their helpful and encouraging feedback. We are glad to know that the reviewers found our work to be comprehensive and valuable to the research community (R-x1j9, R-KPqo, R-bLVp, and R-EdSm), providing a better understanding of the SOTA and holding promise to advance the development of new better 2D-3D bone reconstruction methods (R-HcgH). As the reviewer R-vLVp noted, our work is a “complete package” with curated data, scripts to extract those data, reference implementation of five key architectures with three transformer backbone additions to one of the architectures resulting in eight methods (four architectures’ code were unavailable in public, only 1 method used publicly accessible data, and the three transformer-based backbone was not done in existing studies), the reference implementation of three algorithms that estimate 34 clinical parameters for three anatomies, benchmarking with commonly used ML metrics and the study of its relationship with clinical metrics, and analysis of various kinds of domain shift and robustness. All of this is open source and available at the GitHub repository (temporary link https://anonymous.4open.science/r/xrayto3D-benchmark-ECF4 for now).

There are important feedback from all the reviewers which we have found to be very helpful in improving our paper. We are revising the paper to update the parts of the paper some reviewers kindly highlighted to be a bit unclear, replacing missing “XX” typos to reflect 8 methods on 4 anatomies using 6 datasets, and expanding the discussion/conclusion. Thanks to the comments from the reviewers, we have realized the need to expand and further clarify the relationship between dice and clinical parameters. We are also performing some experiments to address some of the comments.

Most of our responses will be posted as replies to individual relevant comments for easier reading.

---

> ### Author Response · Authors · 2023-08-27
> **Study of Dice score (DSC) and clinical parameters relationship (1/4)**
>
> Thanks to the comments from the reviewers, we have realized that the initial paper did not provide adequate analysis and discussion on the relationship between the DSC and clinical parameters, including an inadequate explanation of Figures 5 and 6. We address this in this comment and will update the manuscript to reflect this response. We would also like to emphasize that this is one of many other important contributions of the paper. Our goal here is to provide the ML community with some quantitative analysis and representative examples with which we hope more people start becoming aware and have tools to explore these when proposing new methods.

---

> > ### Author Response · Authors · 2023-08-27
> > **DICE score (DSC) vs. clinical parameters quantitative results analysis (2/4)**
> >
> > In Figure 5, our initial writing gave the message that better DSC, in general, gives better estimation, but unfortunately, it missed some important details and might have been potentially misleading. We can see in Figure 5 that many parameters where better mean DSC across the whole dataset is not improving the clinical parameter estimation. For example,
> >
> > *First row*: In the fourth and fifth columns OneDConcat has worse DSC by > 5 against UNet, but almost the same PT_R and better IS_L. Similar results can be seen on the 3rd from the right column when comparing OneDConcat against MultiScale2DPermuteConcat or SwinUNETR.
> >
> > *Second row*: Similar observation as above with the DSC difference of > 2 for other pairs of methods in columns 3, 4, and 5.
> >
> > *Third row*: The third column has NSA not much correlated to DSC when DSC is higher than 90. Results similar to the ones stated above for the first two rows can be observed in all the columns from the fourth, with +ve correlation fit in the third column from the right.
> >
> > We provide below some qualitative analysis which will provide some insights into how DSC cannot be related to the parameters monotonically and potential clinical implications.

---

> > ### Author Response · Authors · 2023-08-27
> > **DSC vs. clinical parameters: representative qualitative analysis (3/4)**
> >
> > We provide some cases/scenarios from our datasets and results for each of the three anatomies where DSC and clinical metrics diverge.
> >
> > Figure link https://anonymous.4open.science/r/xrayto3D-benchmark-ECF4/docs/images/clinical_vs_dice.png
> >
> > *Neck-Shaft Angle (NSA)*: As shown in the first row of the figure above, we find that improved DSC is no guarantee that NSA, one of the important parameters for characterizing the proximal femur, is also estimated well. NSA is known to play a role in the diagnosis of various pathologies such as acetabular impingement, hip dysplasia, osteoarthritis, and risk of hip fractures as well as an important parameter in the choice of the implant in Total Hip Arthroplasty [1,2,3].
> >
> > 1. Ripamonti, C., L. Lisi, and M. Avella. "Femoral neck shaft angle width is associated with hip-fracture risk in males but not independently of femoral neck bone density." The British journal of radiology 87.1037 (2014): 20130358.
> >
> > 2. Shoji, Takeshi, et al. "The influence of stem offset and neck shaft angles on the range of motion in total hip arthroplasty." International orthopaedics 40 (2016): 245-253.
> >
> > 3. Mills, H. J., J. G. Horne, and G. L. Purdie. "The relationship between proximal femoral anatomy and osteoarthrosis of the hip." Clinical Orthopaedics and Related Research (1976-2007) 288 (1993): 205-208.
> >
> > *Anterior Superior Illiac Spine {ASIS)*: As shown in the second row of the figure above, the model prediction with similar or higher DSC may not mean a lower clinical estimation error. We see that the model prediction with higher DSC has worse reconstruction performance in the right Anterior Superior iliac spine (ASIS). The error in reconstruction in the ASIS is compensated for by better reconstruction in the posterior ilium region. Since ASIS is used to define the Anterior Pelvic Plane(APP) that defines the anatomical reference frame of the pelvis [1], this error will result in erroneous performance in downstream tasks such as biomechanical evaluation [2], statistical modelling[3] etc.
> >
> > 1. Wu, Ge, et al. "ISB recommendation on definitions of joint coordinate system of various joints for the reporting of human joint motion—part I: ankle, hip, and spine." Journal of biomechanics 35.4 (2002): 543-548.
> >
> > 2. Scheys, Lennart, et al. "Level of subject-specific detail in musculoskeletal models affects hip moment arm length calculation during gait in pediatric subjects with increased femoral anteversion." Journal of biomechanics 44.7 (2011): 1346-1353.
> >
> > 3. von Cramon‐Taubadel, Noreen, Brenda C. Frazier, and Marta Mirazon Lahr. "The problem of assessing landmark error in geometric morphometrics: theory, methods, and modifications." American Journal of Physical Anthropology: The Official Publication of the American Association of Physical Anthropologists 134.1 (2007): 24-35.
> >
> > *Vertebra Canal Length (VCL)*: As seen in the third row of the figure above, similar or higher DSC can result in very different reconstructions in terms of anatomical plausibility as well as clinical parameter estimation. Accurate reconstruction of the vertebra canal is important since the spinal cord passes through it and the error in its reconstruction may result in over or underestimation of the safe zone during pedicle screw fixation surgery [1].
> >
> > 1. Lien, Shiu-Bii, Nien-Hsien Liou, and Shing-Sheng Wu. "Analysis of anatomic morphometry of the pedicles and the safe zone for through-pedicle procedures in the thoracic and lumbar spine." European Spine Journal 16 (2007): 1215-1222.

---

> > ### Author Response · Authors · 2023-08-27
> > **Tolerance levels, actual impact clinical implications, and our contribution (4/4)**
> >
> > The comprehensive study of every single clinical metric, the accuracy of their estimation and relationship to all image-based segmentation metrics based on the tolerance levels, and their impact on the clinical outcome of a patient are out of the scope of the current study since it warrants a self-contained separate (and very interesting) study (or even studies) in itself. However, we believe that our experiments, analysis, and illustrations provide enough evidence that these points are important and lacking in our current ML literature.
> >
> > The changes in clinical parameter estimation error may be within the tolerance limit or outside when compared against DSC. In either case, what our study shows is that our efforts must not be focused only on the development of new methods by solely comparing on image-based metrics such as DSC. If having 90 DSC vs. 95 DSC has little impact on clinical decisions, it might be better to reduce complexity or other relevant parameters (robustness, domain adaptation etc.) rather than trying to improve the DSC.
> >
> > Finally, we would like to emphasize that our motivation/hypothesis behind the addition of clinical metrics was not necessarily to overtly criticize or de-emphasize the use of generalized image-based metrics such as DSC, HD, and ASD but to augment and bring the clinically relevant downstream-task-specific metrics to the attention of the ML community and ultimately encourage optimization for these metrics.

---

### Comment · Area_Chair_QT7p · 2023-08-29
**Reviewers, please check the rebuttal and update your scores**

Dear Reviewers,

The author-reviewer discussion period will end on Aug. 30th. Please check if the authors' rebuttal has addressed your concerns. If not, you can ask the authors for further clarification. Finally, please remember to update your scores if necessary.

Sincerely,

Area Chair

---

### Author Response · Authors · 2023-08-31
**Revised Paper Upload**

Thanks to all the comments from the reviewers which helped us improve the paper. We have now uploaded the revised paper and have removed typos and a few grammar issues highlighted by reviewers. For easy reading, the additions/revisions are coloured in blue text and deletions are removed.

The summary of the major changes in the revised paper are:

*Introduction Section*
- Figure 1 is updated; Old Figure 2 in the original submission moved to Supplementary Material.
- Contributions: clarified the fact that our framework is not limited to encoder-decoder architecture and can be used for any other biplanar X-rays to 3D bone reconstruction methods. The last contribution point about the relationship between DSC and clinical metrics is expanded

*Benchmarking Framework Section*
- Hyperparameter tuning paragraph brought from the discussion section and revised with a clearer description

*Results Section*
- Disaggregated Reporting of Metrics revised
- Reporting of Clinically Relevant Metrics added with expanded presentation and addition of qualitative analysis along with a new Figure (5).
- The previous figure of DSC vs. clinical parameter of a single sample moved to supplementary material, as the new presentation (point above) makes it clearer.
- Misalignment Results were added with results from the additional experiment done.

*Discussion, Limitations, and Future Work*
- Thoroughly revised and expanded.

---

### Decision · Program_Chairs · 2023-09-22

**Decision:**

Accept (Poster)

**Comment:**

This work presented an open benchmark for 3D bone shape reconstruction from biplanar X-ray. Due to a much higher accessibility of X-ray scanners than CT in low-income countries, the task has important clinical applications. It collected six datasets and benchmarked eight encoder-decoder architectures. The presentation is overall clear with minor issues in missing implementation details or typos.

This manuscript received five reviews. All are positive (i.e., a score of 6 or higher). The rebuttal addressed most concerns raised in the initial reviews. The pros and cons (after rebuttal) identified by the reviewers are as follows.

Pros:
1.	A comprehensive open benchmark with six datasets and eight state-of-the-art algorithms.
2.	The algorithms are evaluated with not only standard metrics, such as the Dice score and the Hausdorff distance, but also clinical-relevant metrics, highlighting the potential impact in real clinical practice.
3.	The use of evaluation under domain shift (training on same anatomy but with a different CT data source) adds to the robustness of their findings.

Cons:
1.	The benchmark algorithms are limited to encoder-decoder architectures; however, the platform is open and can be extended to other algorithms.